# A mediator of OsbZIP46 deactivation and degradation negatively regulates seed dormancy in rice

Naihui Guo [1,2,5], Shengjia Tang[1,5], Yakun Wang[1,3], Wei Chen[1], Ruihu An[1], Zongliang Ren[1], Shikai Hu[1], Shaoqing Tang[1], Xiangjin Wei [1], Gaoneng Shao[1], Guiai Jiao[1], Lihong Xie[1], Ling Wang[1], Ying Chen [1], Fengli Zhao [1], Zhonghua Sheng [1,4] ✉ & Peisong Hu [1,2] ✉

Preharvest sprouting (PHS) is a deleterious phenotype that occurs frequently in rice-growing regions where the temperature and precipitation are high. It negatively affects yield, quality, and downstream grain processing. Seed dormancy is a trait related to PHS. Longer seed dormancy is preferred for rice production as it can prevent PHS. Here, we map QTLs associated with rice seed dormancy and clone *Seed Dormancy 3.1* (*SDR3.1*) underlying one major QTL. *SDR3.1* encodes a mediator of OsbZIP46 deactivation and degradation (MODD). We show that SDR3.1 negatively regulates seed dormancy by inhibiting the transcriptional activity of *ABIs*. In addition, we reveal two critical amino acids of *SDR3.1* that are critical for the differences in seed dormancy between the *Xian/indica* and *Geng/japonica* cultivars. Further, *SDR3.1* has been artificially selected during rice domestication. We propose a two-line model for the process of rice seed dormancy domestication from wild rice to modern cultivars. We believe the candidate gene and germplasm studied in this study would be beneficial for the genetic improvement of rice seed dormancy.

Crop domestication is necessary and fundamental for historical development. The food surplus brought about by domestication has promoted the development of human civilization through urbanization and the evolution of the nation state[1]. In turn, the diverse cultural preferences of different historical periods have determined the direction of crop domestication that can meet the particular needs of humans[2–5]. Rice is one of the oldest domesticated crops, and is cultivated approximately 11,000 years ago[6]. Over time, rice has become the main source of energy for more than half of the world's population[7]. During the domestication of wild rice, many physiological traits and morphological features were considerably altered. These transformations include changes in awn development[8–10], seed shattering[11–13], grain quality[6], plant architecture[14–17], grain size[18], heading date[19], panicle size[20], and seed dormancy[21].

An important goal of rice domestication is to increase yield to meet demand and overcome unfavorable environmental conditions. Yield is usually affected by PHS, which generally occurs in high-temperature and high-humidity environments in the middle and lower reaches of the Yangtze River. PHS can cause enormous economic losses in crop production, including declines in seed yield and quality[22]. Longer seed dormancy can prevent PHS, thereby reducing losses and increasing yield. Unfortunately, during domestication, rice seed dormancy is largely lost[23]. Therefore, it is necessary to identify rice cultivars that have

[1]State Key Laboratory of Rice Biological Breeding/Key Laboratory of Rice Biology and Breeding, Ministry of Agriculture/China National Rice Improvement Centre/China National Rice Research Institute, Hangzhou 310006, P. R. China. [2]Rice Research Institute, Shenyang Agricultural University, Shenyang 110866, P. R. China. [3]National Nanfan Research Academy (Sanya), Chinese Academy of Agricultural Sciences, Sanya 572024, P. R. China. [4]Jiangxi Early-season Rice Research Center, Pingxiang, Jiangxi Province 337000, P. R. China. [5]These authors contributed equally: Naihui Guo, Shengjia Tang. ✉e-mail: shengzhonghua@caas.cn; peisonghu@126.com

longer seed dormancy and clone seed dormancy genes to improve rice yield.

Seed dormancy is a complex trait, and many dormancy-related QTLs have been identified in rice;[24–28] however, few of these QTLs have been cloned[21,29–31]. The cloned seed dormancy genes are associated with either abscisic acid (ABA) or gibberellin (GA), two key hormones that regulate seed dormancy[32]. *Sdr4* is the first gene cloned from the mapping population constructed from the *japonica* cultivar Nipponbare and the *indica* cultivar Kasalath[21]. *Sdr4* encodes an unknown function protein, and its regulation of seed dormancy depends on *VP1*. Only *Sdr4-n* occurs in *japonica* cultivars, whereas both *Sdr4-n* and *Sdr4-k* occur in *indica* cultivars, which seems to indicate that *Sdr4* is involved in the rice seed dormancy domestication[21]. *qSD7-1* is cloned from the weedy rice cultivar SS18-2 and encodes a *basic helix-loop-helix* protein. It increases the ABA content in seeds by upregulating the expression of ABA synthesis genes[29]. In addition, *qSD7-1* interacts with *VP1* and *OsC1* to increase the seeds sensitivity to ABA[33]. *qSD1-2* is isolated from SS18-2. It regulates seed dormancy via the GA pathway[30]. Recently, *SD6*, which is cloned from Kasalath, together with *ICE2* antagonistically balanced the expression of ABA metabolic genes to control rice seed dormancy by responding to temperature signals[31]. However, if seed dormancy is too persistent, it is difficult to sprout, which increases the cost of timely farming; conversely, if dormancy is too weak, PHS will occur. Thus, it is necessary to appropriately regulate seed dormancy by identifying additional seed dormancy genes.

ABA is an important hormone that regulates seed dormancy. Its signal transduction involves many genes, such as the transcription factors *ABI3* and *ABI5*. These genes positively regulate the transcription factors involved in rice seed dormancy[34–36]. Other ABA signaling response factors, such as e*mbryonic abundant protein* (*EM1*)[37], *late embryogenesis abundant protein* (*LEA3*)[38], and the *mediator of OsbZIP46 deactivation and degradation* (*MODD*), are often involved in regulating drought resistance in rice[39]. *MODD*, which is homologous to the *Arabidopsis thaliana* ABI5-binding protein AFP, negatively regulates drought tolerance in rice by regulating OsbZIP46 activity and stability[39]. However, whether it regulates rice seed dormancy remains unknown.

In this work, we clone the major QTL *qSDR3.1* associated with seed dormancy. We show that the underlying gene *SDR3.1*, encoding a mediator of OsbZIP46 deactivation and degradation (MODD), represses rice seed dormancy by inhibiting the transcriptional activity of *ABI*s. Population genetics analysis reveals that *SDR3.1* has been intensively selected during domestication and breeding, especially for *Geng/japonica* cultivars. We demonstrate the usefulness of *SDR3.1* by introducing it into a rice restorer line cultivar for hybrid seed production.

## Results

### Phenotypic characterization of the two parental lines

ZH11 is a *japonica* cultivar with a high postharvest germination rate (83.00%), whereas Mengjialaxiaoli (MJLXL) is an *Aus* cultivar with a low post-harvest germination rate (4.11%; Supplementary Fig. 1a, b). The germination rates were measured immediately at 35 days after heading (DAH). Three months after harvest, the germination rates of ZH11 and MJLXL both reached to approximately 80% (Supplementary Fig. 1c, d). These results demonstrate that MJLXL is a cultivar with longer seed dormancy and ZH11 is a cultivar with shorter seed dormancy. Therefore, ZH11 and MJLXL were selected as parental lines to develop the mapping population.

### *qSDR3.1* is a major QTL that regulates seed dormancy

To determine the genetic mechanism underlying rice seed dormancy, we used ZH11 as the recipient parent and MJLXL as the donor parent to construct a backcross inbred lines population (Fig. 1a). First, we examined the germination rate of 153 lines derived from the $BC_3F_2$ population, which were selfbred from a single $BC_3F_1$ plant (the genotype of $BC_3F_1$ is shown in Supplementary Data 1). The germination rate of this population exhibited a normal distribution, indicating that seed dormancy was controlled by QTLs (Supplementary Fig. 2a). Then, we identified three QTLs of seed dormancy, designated *qSDR2.1*, *qSDR3.1*, and *qSDR3.2*, which were located on the second and third chromosomes, respectively (Supplementary Fig. 2b). *qSDR3.1* had the highest logarithm of odds value (11.75) and was mapped between markers 3-15 and 3-21 on chromosome 3 (Supplementary Fig. 2c; Supplementary Table 1). Genetic analysis of the $BC_3F_2$ individuals suggested that *qSDR3.1* is an incomplete dominant gene (Supplementary Fig. 2d, e).

The six homozygous recombinant lines (selected from 4146 $BC_4F_2$ individuals) in the QTL region were analyzed for fine mapping (Fig. 1b). The 7-day germination rate phenotype of the recombinant lines ranged from 20.16% to 72.14%. The 7-day germination rate of R2, R4, and R6 were similar to that of MJLXL, and the 7-day germination rate of R1, R3, and R5 were similar to that of ZH11. Finally, the location of *qSDR3.1* was narrowed to a 26.9-kb region between markers Indel27 and Indel29 (Fig. 1b). This region contained 4 candidate genes, namely, *LOC_Os03g11520*, *LOC_Os03g11530*, *LOC_Os03g11540* and *LOC_Os03g11550* (Supplementary Table 2). *LOC_Os03g11520* encode a proteins with unknown function. There was no difference between the parents in *LOC_Os03g11530* and *LOC_Os03g11540*. And there were three bases differences between the two parents in *LOC_Os03g11550*, which created two amino acid differences (Fig. 1c; Supplementary Fig. 3a-b).

In addition, the cDNA of *LOC_Os03g11550* of R3 was consistent with that of ZH11, whereas the cDNA of *LOC_Os03g11550* of R2 and R4 were consistent with that of MJLXL, indicating that the *LOC_Os03g11550* genotypes cosegregated with the seed dormancy phenotypes (Supplementary Fig. 3c-e). According to the Rice Genome Annotation Database (http://rice.uga.edu/, RGAP 7), *LOC_Os03g11550* comprises three exons and two introns. It encodes MODD[39]. *MODD* negatively regulates ABA signaling and drought resistance in rice[39]. Therefore, *LOC_Os03g11550* was selected as the candidate gene of *SDR3.1*.

To further identify *SDR3.1*, six chromosome segment substitution lines from the $BC_3F_3$ population were selected: R15, R32, and R152, which had a high germination rate; R16, R23, and R100, which had a low germination rate. The genotypes of *SDR3.1* of R15, R32, and R152 were the same as that of ZH11, whereas the genotypes of *SDR3.1* of R16, R23, and R100 were the same as that of MJLXL (Fig. 1c). We complemented *SDR3.1* from ZH11 into NIL. The copy numbers of COM-1, COM-2, and COM-3 were 13, 7 and 7, respectively (Fig. 1e). The 7-day germination rate of the $T_2$ generation seeds of the three complementary lines (COM-1: 90.06%; COM-2: 50.95%; COM-3: 52.50%) were significantly higher than that of NIL (19.01%) (Fig. 1d, f). These results demonstrated that *LOC_Os03g11550* is the target gene of *SDR3.1*.

### *SDR3.1* negatively regulates seed dormancy

To clarify the function of *SDR3.1*, we disrupted it in ZH11 via CRISPR/Cas9, and two homozygous mutants, KO-1 and KO-2, were screened through sequencing. These mutants contained a 1-bp frame-shift insertion or deletion in the first exon (Supplementary Fig. 4a, Supplementary Fig. 5). The expression and protein levels of *SDR3.1* in both mutant lines were significantly lower than those of ZH11 (Supplementary Fig. 4b-c). Similarly, the 7-day germination rate of the two mutant lines (KO-1 and KO-2: 9.59% and 17.09%, respectively) were significantly lower than that of ZH11 (82.56%) (Supplementary Fig. 4d-e). Thus, we concluded that *SDR3.1* negatively regulates seed dormancy.

In rice, PHS frequently leads to severe decreases in grain yield and quality. Enhanced seed dormancy is useful for preventing PHS[40]. To determine whether *SDR3.1* regulates PHS, we measured the PHS of

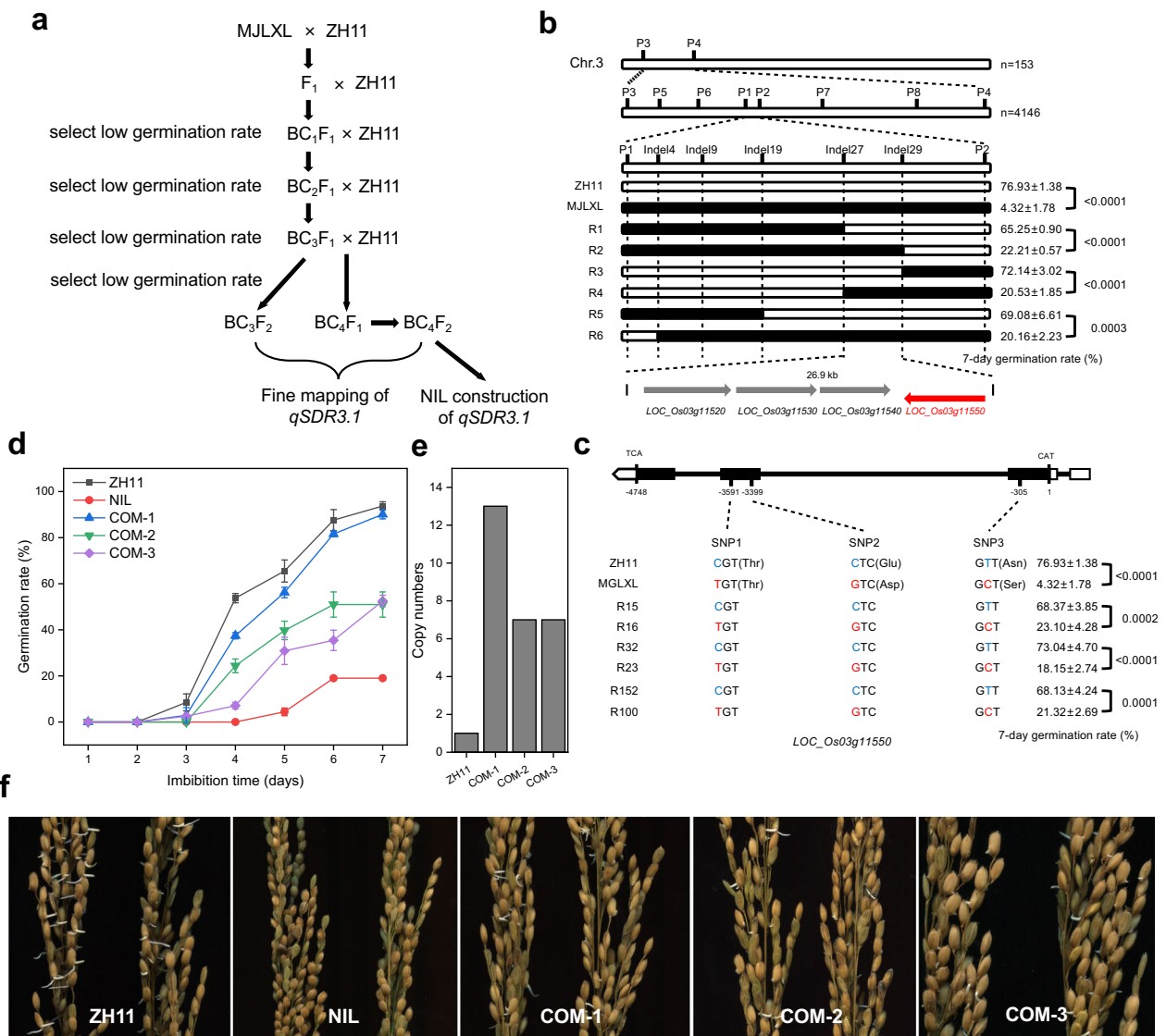

**Fig. 1 | Fine mapping of *SDR3.1*. a** Roadmap of constructing near-isogenic lines and fine-mapping populations. **b** *SDR3.1* was located between markers Indel27 and Indel29 on chromosome 3. **c** Identification of candidate genes of *SDR3.1*. **d** Germination rate of ZH11, the NIL, and three complementary lines 35 DAH (*n* = 3 independent experiments). **e** Copy numbers of *SDR3.1* in ZH11 and three complementary lines. **f** PHS phenotypes of ZH11, the NIL, and three complementary lines 35 DAH. Data are presented as the mean ± SD. Source data are provided as a Source Data file.

ZH11, MJLXL, NIL, and the two mutant lines. On the 6th day, the PHS of NIL and MJLXL significantly decreased by 82.83% and 83.51% compare with that of ZH11, respectively. On the 5th day, the PHS of KO-1 and KO-2 significantly decreased by 48.21% and 47.79% compare with that of ZH11, respectively (Supplementary Fig. 6). Combined with the enhanced seed dormancy in NIL and the two mutant lines (Fig. 1d; Supplementary Fig. 4d, e), indicating that *SDR3.1* regulates PHS.

### Expression and subcellular localization of *SDR3.1*

qRT-PCR showed that *SDR3.1* is widely expressed in 10-day leaf, glume, anther, 5-day root, 20-day root, and 5-day leaf, with the highest transcript level detected in the pistil (Supplementary Fig. 7a). Then, the dynamic transcript levels of *SDR3.1* were investigated during seed development. The transcript began to accumulate 5 days after fertilization and peaked at 15 days (Supplementary Fig. 7a).

To study the biological function of *SDR3.1*, we fused the coding region to the N-terminus of GFP. The empty *GFP* vector was used as a control and mCherry (*LOC_Os11g01330*) was used as a marker. The results showed that the 35 S::SDR3.1-GFP protein was localized in the nucleus (Supplementary Fig. 7b).

### SDR3.1 physically interacts with ABI5 and represses its transcriptional activation

To determine whether *SDR3.1* is a transcription factor, the recombinant vector pGBKT7-*SDR3.1* was transferred into the yeast (*Saccharomyces cerevisiae*) strain Y2HGold. Cells expressing SDR3.1 protein could not grow on the systematic tri-deficient medium (Fig. 2a), indicating that SDR3.1 was not a transcription factor.

Because MODD is homologous to the ABI5-binding protein AFP in *Arabidopsis thaliana*[39], we speculated that SDR3.1 could interact with ABI5 in rice. To test this hypothesis, we performed a yeast two-hybrid (Y2H) assay. The results showed that SDR3.1 interacted with ABI5 (Fig. 2b), which was further confirmed by the bimolecular fluorescence complementation (BiFC) assay (Fig. 2c). Moreover, a protein–protein pull-down assay showed that GST-SDR3.1 bound His-ABI5 (Fig. 2d), indicating that SDR3.1 interacted with ABI5 in vitro. In summary, SDR3.1 could interact with ABI5, implying that *SDR3.1* mediates the ABA response to regulate seed dormancy.

To identify the protein sites necessary for interaction with ABI5, we divided SDR3.1 into three segments according to the

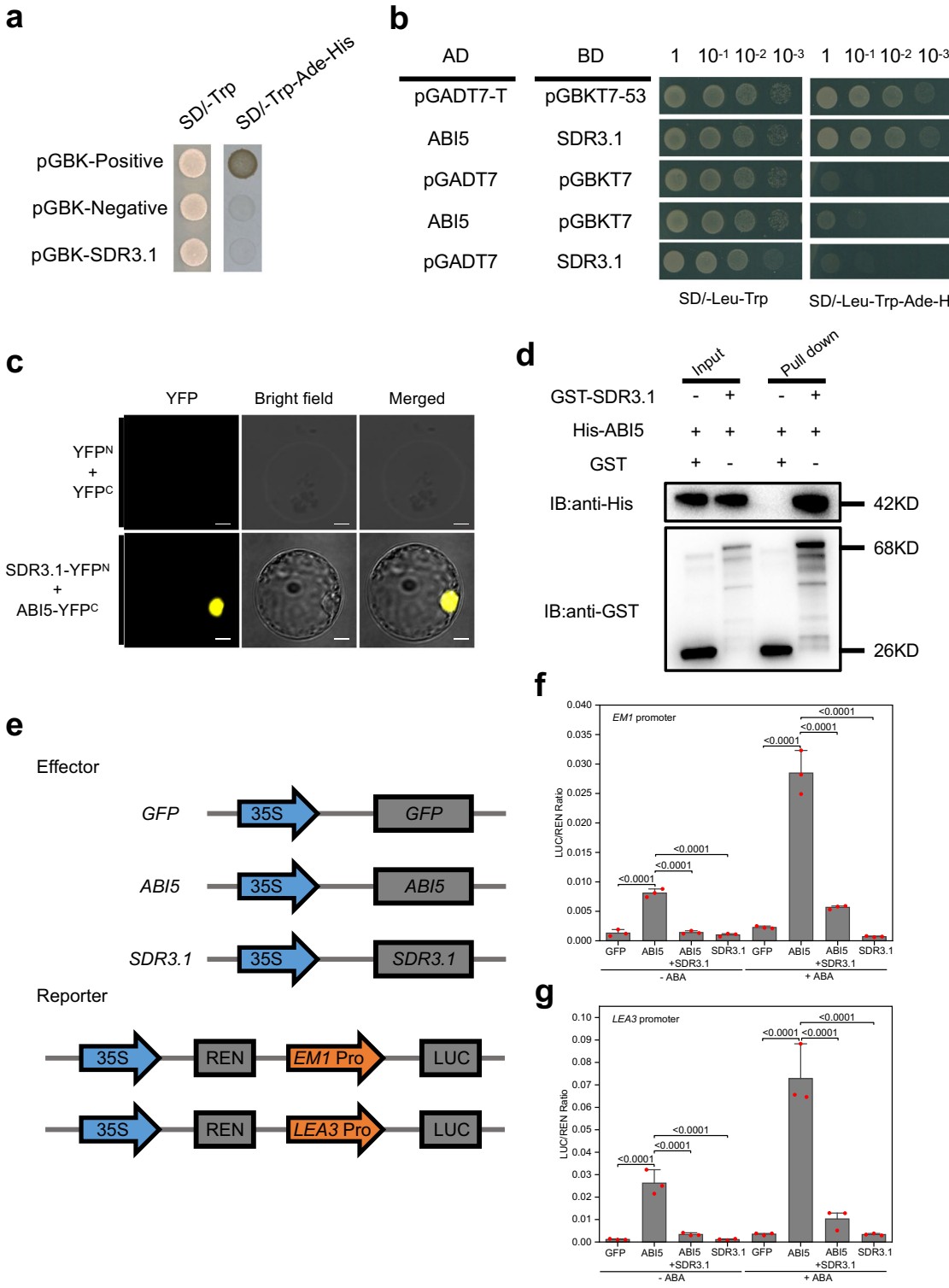

**Fig. 2 | SDR3.1 physically interacts with ABI5. a** Verification of SDR3.1 transcriptional activation; pGBK-Negative, negative control; pGBK-Positive, positive control. **b** The yeast two-hybrid assay showed that SDR3.1 interacted with ABI5. pGADT7-T and pGBKT7-53 were used as positive controls; empty vectors were used as negative controls. **c** Bimolecular fluorescence complementation analysis. Fluorescence was observed only in the nuclei of protoplast, which resulted from complementation of the N-terminal region of YFP fused with *SDR3.1* (SDR3.1-YFP^N) with the C-terminal region of YFP fused with *ABI5* (ABI5-YFP^C). Bar = 10 μm. The experiments were replicated 3 times with similar results. **d** In vitro pull-down assays

of SDR3.1 with ABI5. Affinity-purified His-ABI5 was incubated with GST-SDR3.1. Pull-down proteins were subjected to immunoblotting with anti-His antibodies. The experiments were replicated 3 times with similar results. **e** Schematic of the effectors and reporter used in the transient transactivation assays. **f** SDR3.1 repressed ABI5 to activate *EM1* in response to 5 μM ABA. **g** SDR3.1 repressed ABI5 to activate *LEA3* in response to 5 μM ABA. In **f**, **g** *n* = 3 independent experiments. REN, Renilla LUC. Data are presented as the mean values ± SD, and *P* values are indicated by two-tailed Student's *t* test. Source data are provided as a Source Data file.

NCBI conservative domains database (https://www.ncbi.nlm.nih.gov/Structure/cdd/wrpsb.cgi), namely, the N-terminus containing the EAR domain (BD-SDR3.1-N), the M-terminus containing the NINJA_B domain (BD-SDR3.1-M), and the C-terminus containing the Jas domain (BD-SDR3.1-C; Supplementary Fig. 8a). Deletion of the N- and M-terminus residues did not affect the interaction of SDR3.1 with ABI5; however, deletion of the C-terminus fragment of SDR3.1 abolished its interaction with ABI5 (Supplementary Fig. 8b). Therefore, the C-terminus of SDR3.1 was necessary for the SDR3.1-ABI5 interaction.

Similarly, we divided *ABI5* into two segments; an N-terminus domain (AD-ABI5-N) and a C-terminus bZIP domain (AD-ABI5-C; Supplementary Fig. 8c). Deletion of the C-terminus residue of ABI5 did not affect its interaction with SDR3.1, whereas deletion of the N-terminus fragment abolished the interaction of ABI5 with SDR3.1 (Supplementary Fig. 8d). Therefore, the N-terminus of ABI5 was essential for the SDR3.1-ABI5 interaction.

To clarify the function of *ABI5*, we disrupted it (*LOC_Os01g64000*) in ZH11 via CRISPR/Cas9, and two homozygous mutants, *abi5*-1 and *abi5*-2, were obtained through sequencing. These mutants contained a 1-bp frame-shift insertion or deletion in the first exon (Supplementary Fig. 9a), and the mutants had lower ABI5 expression levels than that of ZH11 (Supplementary Fig. 9b). Moreover, we overexpressed *ABI5* in ZH11 (Supplementary Fig. 9c). The results indicated that the mutant lines (97.92% and 98.33%) had higher germination rate than that of ZH11 (88.75%). In contrast, the germination rate of the overexpressed lines (36.67% and 46.67%) were significantly lower than that of ZH11 (88.75%) (Supplementary Fig. 9d-e). These results indicated that *ABI5* promoted seed dormancy.

Because *SDR3.1* inhibits seed dormancy and *ABI5* promotes seed dormancy, we investigated whether the SDR3.1 protein inhibits the transcriptional activation of *ABI5*. Therefore, identifying *ABI5* target genes was necessary. In Arabidopsis, the *EMBRYONIC ABUNDANT PROTEIN1* (*EM1*) is a target of *ABI5*[41], but *EM1* has not been reported as a target gene of *ABI5* in rice. Therefore, we used a yeast one-hybrid (Y1H) assay to confirm that ABI5 could bind to the promoter of *EM1* in rice (Supplementary Fig. 10a), and we narrowed the binding region within 200 bp (Supplementary Fig. 10b). In addition, we used an electrophoretic mobility shift assay (EMSA) to confirm the interaction between ABI5 and the promoter of *EM1* (Supplementary Fig. 10c). To identify additional downstream target genes of *ABI5*, we used Cleavage Under Targets and Tagmentation (cut&Tag) technology combined with Y1H and EMSA to verify that ABI5 could bind to the promoter of *LEA3* (Supplementary Fig. 10d-f).

Furthermore, we disrupted *EM1* (*LOC_Os05g28210*) and *LEA3* (*LOC_Os05g46480*) in ZH11 via CRISPR/Cas9, and four homozygous mutants, namely, *em1*-1, *em1*-2, *lea3*-1 and *lea3*-2 were obtained through sequencing. These mutants contained a 32-bp or 28-bp frame-shift deletion in the first exon of *EM1* and an 89-bp frame-shift deletion or 1-bp frame-shift insertion in the second exon of *LEA3* (Supplementary Fig. 11a-b). The *em1*/*lea3* plants had lower gene expression levels than that of ZH11 (Supplementary Fig. 11c). Moreover, we overexpressed *EM1* and *LEA3* in ZH11, respectively (Supplementary Fig. 11c). The mutant lines had higher germination rate than that of ZH11. In contrast, the germination rate of the overexpressed lines were significantly lower than that of ZH11 (Supplementary Fig. 11d-e). These results indicated that *EM1* and *LEA3* are important downstream genes involved in the seed dormancy regulation pathway.

Since *EM1*-*ABI5* and *LEA3*-*ABI5* interact in rice, we fused the *LUC* gene to the *EM1* and *LEA3* promoters (Fig. 2e). In trans, we expressed *SDR3.1*, *ABI5*, and *GFP* with the CaMV 35 S promoter (Fig. 2e). The *EM1* and *LEA3* reporters were significantly activated by the *ABI5* effector, when compared with the *GFP* effector (Fig. 2f-g). However, the coexpression of SDR3.1 with ABI5 disrupted the ABI5-activated *LUC* expression (Fig. 2f-g). These results indicated that SDR3.1 repressed the transcriptional activity of ABI5.

To verify the relationship between *SDR3.1* and *ABI5*, we constructed a double mutant in ZH11 (Supplementary Fig. 12a-b). The germination rate of the double mutants had no significantly difference compare with that of ZH11 (Supplementary Fig. 12c-d). These results further verified that SDR3.1 repressed the transcriptional activity of ABI5.

### SDR3.1 negatively regulates ABA signaling

Because *ABI5* is the central factor corresponding to ABA signaling, we examined the expression levels of ABA response genes in ZH11, MJLXL and transgenic plants. In addition to *OsEM1* and *OsLEA3*, we also detected the expression levels of three genes that are related to ABA regulation (*OsbZIP72*, *OsRab16A* and *OsTRAB1*). The results showed that the expressions were suppressed significantly by *SDR3.1* (Fig. 3a-e). However, the expression levels of *OsbZIP72*, *OsRab16A* and *OsTRAB1* were not activated by *ABI5* (Fig. 3c-e). Therefore, *SDR3.1* might regulate the response to ABA signals through more than just *ABI5*. Based on these findings, we tested whether *SDR3.1* inhibits another seed dormancy-related transcription factor, *OsABI3*. The results showed that SDR3.1 inhibited the transcriptional activity of OsABI3 (Supplementary Fig. 13a-b).

In addition, we examined the ABA and GA contents in seeds, and the results showed that *SDR3.1* regulated seed dormancy not by regulating ABA and GA contents (Fig. 3f; Supplementary Fig. 13c-f). Then, we examined the sensitivity of ZH11 and the two mutant lines to ABA after breaking seed dormancy. As expected, under normal conditions, there was no significant differences in germination rate between ZH11 and the two mutant lines (Fig. 3g-i). However, after 5 µM or 10 µM ABA treatment, the germination rate of the two mutant lines were significantly lower than that of ZH11 (Fig. 3g-i). Therefore, the two mutant lines were more sensitive to ABA. The above results indicated that SDR3.1 negatively regulated ABA signaling by repressing ABIs.

### Evolutionary analysis of *SDR3.1*

The *SDR3.1* encoded protein contains 401 amino acids. Homologous proteins were detected by BLAST searches of the UniProt database (https://www.uniprot.org/). Species with more than 50% similarity to SDR3.1 protein sequences were selected for multiple sequence alignment, and a phylogenetic tree was constructed using the neighborhood connection method. Homologs of SDR3.1 were identified only in monocotyledons, including *Setaria viridis*, *Setaria italica*, *Panicum miliaceum*, *Dichanthelium oligosanthes*, *Sorghum bicolor*, *Zea mays*, and *Eragrostis curvula*. The highest identities observed of SDR3.1 protein sequences were from the wild rice species *Oryza glumipatula* (99.8%), *Oryza barthii* (99.5%), *Oryza meridionalis* (96.8%) and *Oryza punctata* (93.5%) (Supplementary Fig. 14). These results implied that SDR3.1 performs an irreplaceable function in monocotyledons.

### Natural variations of *SDR3.1* associated with differences in seed dormancy

Given that seed dormancy is an ancient trait of wild rice[13] and that most modern rice cultivars exhibit shorter seed dormancy. We analyzed the polymorphisms of *SDR3.1* in the 3 K RG dataset (www.rmbreeding.cn/index.php). Using five single-nucleotide polymorphisms (SNPs) in the coding sequence, we detected five major haplotypes (Hap) of *SDR3.1* in 2584 accessions in the 3 K RG dataset (only haplotypes with more than 100 accessions were selected) (Fig. 4b). These were divided mainly into *Xian*/*indica* (58.6%), *Geng*/*japonica* (29.8%), and *Aus* (6.9%) cultivars. Further analysis revealed that *Xian*/*indica* was mainly Hap1 (52.5%) and Hap2 (44.4%), *Geng*/*japonica* was mainly Hap3 (51.5%) and Hap4 (48.5%), and *Aus* was mainly Hap5 (69.1%), Hap2 (18.0%) and Hap1 (12.4%) (Supplementary Fig. 15). Further, we analyzed 2356 cultivars that the exact geographic information was available. Each haplotype was distributed in the region where rice was grown (Fig. 4a), and the

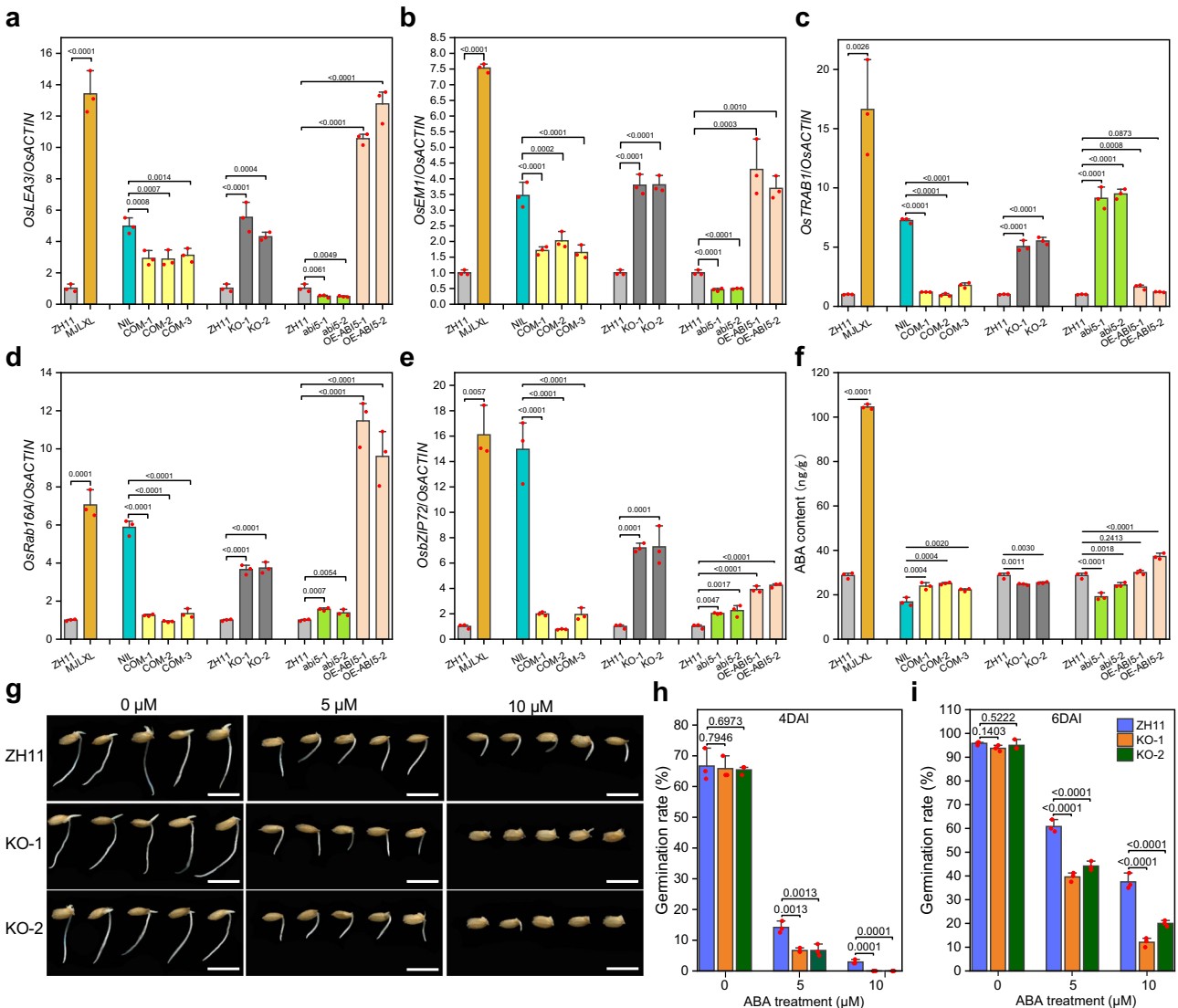

**Fig. 3 | SDR3.1 negatively regulates ABA signaling. a–e** The relative expression levels of ABA-responsive genes between ZH11 and MJLXL, among the NIL and three COM lines, among ZH11 and two *SDR3.1* knockout lines, among ZH11, two *ABI5* knockout lines and two OE lines. **f** ABA contents between ZH11 and MJLXL, among the NIL and three COM lines; among ZH11 and two *SDR3.1* knockout lines; and among ZH11, two *ABI5* knockout lines and two OE lines. **g** Germination performance of two *SDR3.1* knockout lines and ZH11 seeds treated with 0, 5, or 10 μM ABA at

4 days after imbibition (DAI). Bar = 1 cm. **h** Germination rate of two *SDR3.1* knockout lines and ZH11 seeds treated with 0, 5, or 10 μM ABA at 4 DAI. **i** Germination rate of two *SDR3.1* knockout lines and ZH11 seeds treated with 0, 5, or 10 μM ABA at 6 DAI. In **a–f, h, i** n = 3 independent experiments. Data are presented as the mean values ± SD, and *P* values are indicated by two-tailed Student's *t* test. Source data are provided as a Source Data file.

Shannon evenness (E_H) value of *SDR3.1* was 0.192[42], indicating that the genetic diversity of *SDR3.1* was relatively low among cultivars.

We collected 291 rice accessions to determine the association between genotypes and seed dormancy phenotypes. The average germination rate of Hap3 (67.68%) and Hap4 (59.73%) were the highest, followed by those of Hap1 (31.87%) and Hap2 (24.28%), and the average germination rate of Hap5 (4.95%) was the lowest (Fig. 4c; Supplementary Figs. 16-20). Hap1 was mainly *Xian/indica* and *Aus*, accounting for 63.77% and 24.64%, respectively. Hap2 was mainly *Xian/indica* and *Aus*, accounting for 45.78% and 44.58%, respectively. Hap3 and Hap4 were dominated by *Geng/japonica*, accounting for 77.56% and 97.22%, respectively, while 98.28% of the Hap5 was *Aus* (Fig. 4d). These results suggested that the function of *SDR3.1* differs between the *Xian/indica* and *Geng/japonica* cultivars.

Due to the similar germination rate between Hap1 and Hap2, as well as between Hap3 and Hap4, we speculated that different SNPs may have caused synonymous mutations. We subsequently

analyzed the differences in amino acids at the five different SNP sites. We found only two of five SNPs (SNP-3399 and -305 in Fig. 4b) changed the amino acids; the other three SNPs (SNP-3591, -348 and -75 in Fig. 4b) did not change the amino acids. Therefore, the amino acid sequences of *SDR3.1* of Hap1 and Hap2 were the same (Ser[102]+Asp[156]), the amino acid sequences of *SDR3.1* of Hap3 and Hap4 were also the same (Asn[102]+Glu[156]), and Hap5 was Ser[102]+Glu[156]. Among the combinations tested, Asn[102]+Glu[156] was associated with the highest germination rate (64.13%), followed by Ser[102]+Asp[156] (27.72%), and Ser[102]+Glu[156] was associated with the lowest germination rate (4.95%) (Fig. 4e). The combination of Asn[102]+Glu[156] was mostly appear in the *Geng/japonica* cultivar (85.19%), the combination of Ser[102]+Asp[156] was mostly appear in the *Xian/indica* cultivar (53.95%), and the combination of Ser[102]+Glu[156] was mostly appear in the *Aus* cultivar (98.28%) (Fig. 4f). These results implied that *Aus* had longest seed dormancy, followed by *Xian/indica*, while *Geng/japonica* had the shortest seed dormancy.

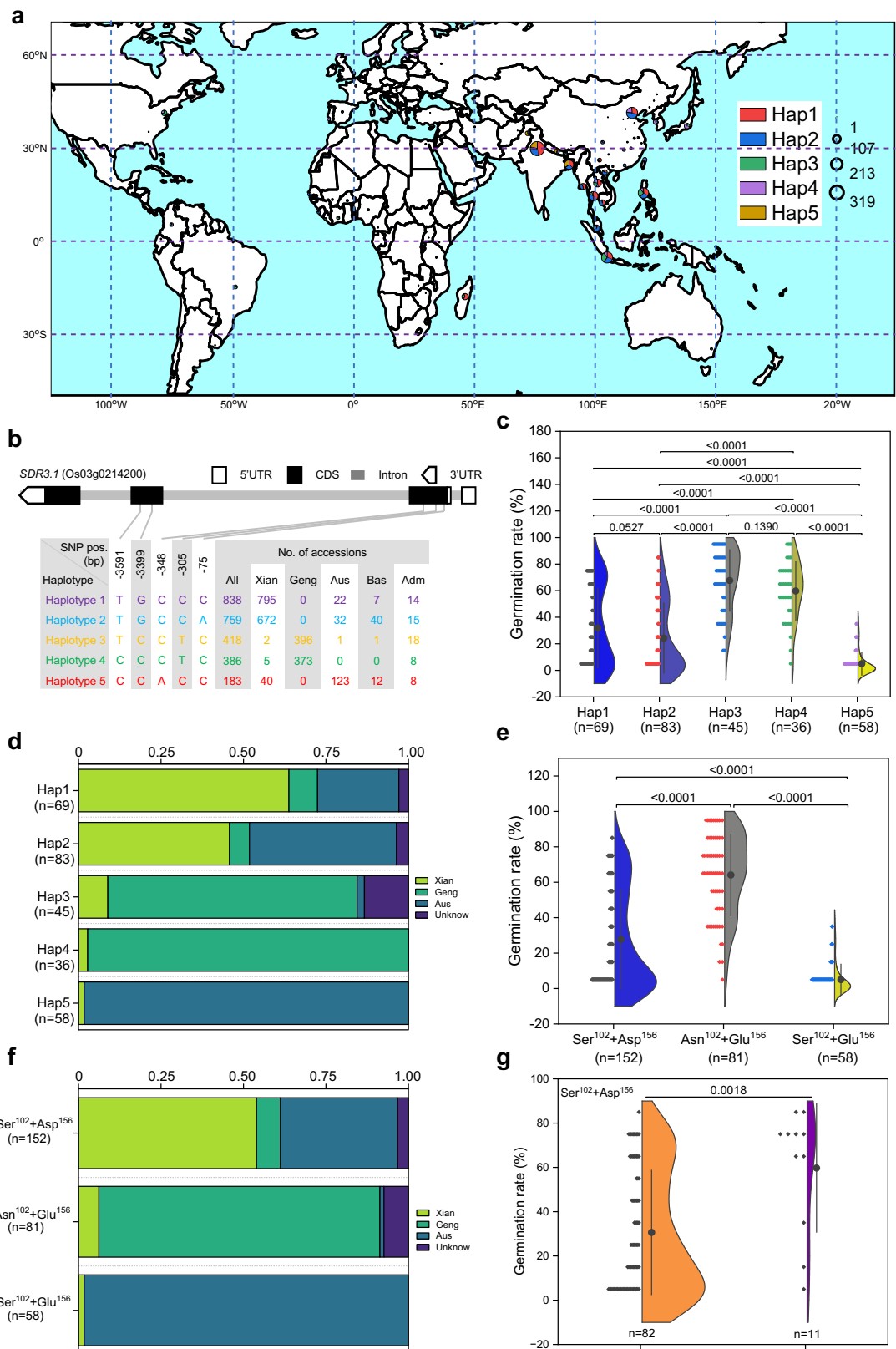

**Fig. 4 | Natural variation in *SDR3.1* is associated with differences in seed dormancy. a** Geographic distribution of the five haplotypes among 2356 accessions of 3 K RGs. N: North latitude; S: South latitude; W: West longitude; E: East longitude. **b** Haplotypes of *SDR3.1* in 2584 accessions of 3 K RG according to five SNPs in the coding sequences region. **c** The 7-day germination rate of the five haplotypes (291 rice accessions) 35 DAH. The short horizontal lines represent the number of cultivars. **d** Rice subspecies distribution of each haplotypes. **e** The 7-day germination rate of the three combinations (291 rice accessions) 35 DAH. The short horizontal lines represent the number of cultivars. **f** Rice subspecies distribution of each amino acid sequences. **g** The differences of germination rate between the *Xian/indica* and *Geng/japonica* in the combination of $Ser^{102} + Asp^{156}$. The short horizontal lines represent the number of cultivars. In **c, e** Data are presented as the mean ± SD, and *P* values are indicated by two-tailed Student's *t* test. Source data are provided as a Source Data file.

The genetic effects of different alleles are strongly affected by the population structure as Asian cultivated rice exhibit dramatic differences in *Xian/indica* and *Geng/japonica* rice cultivars. Therefore, we compared seed dormancy between the *Xian/indica* and *Geng/japonica* cultivars in both Hap1 and Hap2 with Ser[102]+Asp[156], and the results showed that the germination rate of *Geng/japonica* was significantly higher than that of *Xian/indica* (Fig. 4g), indicating that the genetic effect of dormancy was influenced by population structure.

To investigate whether the effect of *SDR3.1* on seed dormancy was related to its expression levels, we measured its expression levels in some rice cultivars from each haplotype. There was no significant correlation between the germination rate and the expression levels of each haplotype (Supplementary Fig. 21a-e). There were also no significant differences in expression levels among the five haplotypes, except between haplotype 4 and haplotype 5 or between haplotype 1 and haplotype 4 (Supplementary Fig. 21f). These results indicated that the regulation of seed dormancy by *SDR3.1* depended on its protein activity more than its expression levels.

### Conversion of longer dormancy in wild rice to shorter dormancy in *Xian/indica* and *Geng/japonica* requires *SDR3.1* aa[102] and aa[156]

To clarify the evolution of *SDR3.1* in rice, we analyzed the amino acid sequences of the 4 wild rice accessions (Supplementary Fig. 14). The amino acid sequences at positions 102 and 156 were Ser and Glu, respectively, which was consistent with the sequence of Hap5 (Fig. 5a). We subsequently sequenced 28 wild rice accessions and found that most of the amino acid sequences at positions 102 and 156 were Ser and Glu (Fig. 5b). The absolute Tajima values for *Xian/indica* and *Geng/japonica* were both greater than 2, their π values decreased sharply around *SDR3.1*, and *Geng/japonica*'s π values were lower than that of *Xian/indica* (Fig. 5c-d; Supplementary Fig. 22). These findings suggested that *SDR3.1* was selected during rice domestication and was more strongly selected in *Geng/japonica*.

Based on the varying degrees of selective effects between the *Xian/indica* and *Geng/japonica* cultivars, which may have an impact on the germination rate. We investigated the germination rate of some cultivars among different rice subspecies. The results showed that the germination rate was as follows: *Geng/japonica* (such as TD-156, TD-187, TD-198, and TD-229) > *Xian/indica* (such as TD-45, TD-13, TD-92, and TD-104) > *Aus* (such as TD-238 and TD-256) (Supplementary Fig. 23). To determine the molecular functions of aa[102] and aa[156], we constructed effectors of five haplotypes (Supplementary Fig. 24a), cotransformed rice protoplasts with *ABI5*, and detected the LUC activity of *EM1* and *LEA3*. The results showed that Hap3 and Hap4 had the strongest inhibitory effects on *ABI5* transcriptional activity, Hap1 and Hap2 had the second strongest inhibitory effects, and Hap5 had the weakest inhibitory effect (Supplementary Fig. 24b). Furthermore, we constructed a site-directed mutation mutant containing the aa[102] or aa[156] substitutions; namely, we changed the asparagine acid (N) to the serine acid (S) at site 102 in ZH11 and changed the aspartic acid (D) to glutamic acid (E) at site 156 in YK17. The results showed that the germination rate of the mutants were significantly lower than that of ZH11 or YK17 (Fig. 5e-f). Taken together, these results indicated that the conversion of the longer seed dormancy of wild rice to the shorter seed dormancy of *Xian/indica* and *Geng/japonica* required aa[102] or aa[156]: a change from Ser[102] to Asn[102] to produce *Geng/japonica* and a change from Glu[156] to Asp[156] to produce *Xian/indica* (Fig. 5g).

### Seed dormancy improvement of the high-quality rice restorer line

Zhonghui261 (ZH261) is a high-quality restorer line, but its seed dormancy persistence needs to be further improved. Therefore, we introduced *SDR3.1* from MJLXL into ZH261. We obtained BC$_1$F$_1$ populations with 3565 plants by hybridizing MJLXL with ZH261 (Fig. 6a). Then, the plants that were heterozygous for *SDR3.1* were

selected by molecular markers detection, and the single plant named BC$_1$F$_1$-80 with the highest degree of genetic background reversion was selected by sequencing with a rice 40 K gene chip. The progeny of the BC$_1$F$_1$-80 selfing were marker detected to select lines with homozygous alleles of MJLXL at *SDR3.1*, and the single plant named BC$_1$F$_2$-39 with the highest degree of genetic background reversion was subsequently selected by sequencing with a rice 40 K gene chip (Fig. 6b). BC$_1$F$_2$-39 exhibited 88.78% genetic similarity to ZH261 (Supplementary Fig. 25a-c). As expected, BC$_1$F$_2$-39 had significantly lower germination rate than that of ZH261 (Fig. 6d-e), whereas the 1000-grain weight, panicle length, grain length, and grain width (Fig. 6c, f) had no significantly differences than those of ZH261. The seed dormancy of the other two sister lines also increased significantly, indicating that *SDR3.1* indeed increased the seed dormancy of the improved lines (Supplementary Fig. 25d-i).

## Discussion

Rice germplasm resources are a diverse collection of rice varieties that have been conserved for their genetic diversity and potential utility in breeding programs. The seed hull gene *Bh4*[43], the awn development gene *An-1*[8], and the plant architecture gene *PROG1*[14] were cloned from wild rice, and the hypoxia tolerance gene *GF14h* was recently cloned from natural weedy rice[44,45]. However, rice seed dormancy gene cloning is rare[21,46]. In this study, we cloned a major QTL, *SDR3.1*, from the longer seed dormancy cultivar MJLXL. This finding suggested that cloning elite genes by using rice germplasm resources is feasible. During the process of cloning *SDR3.1*, we found that the germination rate of COM-1 was significantly higher than that of COM-2 and COM-3 (Fig. 1d). The copy number results showed that the copy number of COM-1 was almost twice that of COM-2 and COM-3 (Fig. 1e), and copy number variation affects gene function[47]. Therefore, we speculated that the significant differences in the germination rate among COM-1, COM-2 and COM-3 is due to copy number variation.

Seed dormancy and germination are regulated by internal and external signals[48]. Among these signals, the hormone ABA is an important regulator that inhibits seed germination[32]. Consistent with this, most seed dormancy genes are involved in ABA synthesis and ABA signal transduction[21,33,46,49]. *qSDR3.1* participates in the ABA signal transduction pathway to regulate seed dormancy through interaction with *ABI5* (Fig. 2b-d). *ABI5* is a key regulator of ABA signal transduction[50]. The *abi5* mutant has been found to be insensitive to ABA during germination, whereas plants that overexpress *ABI5* are hypersensitive to ABA and exhibit enhanced seed dormancy[34,35]. GA[51], sugar[35], cytokinin[52], the brassinolide (BR) signaling factor *BES1*[53] and the jasmonic acid (JA) signaling inhibitor *JAZ*[54,55] can antagonize the effect of ABA by inhibiting the activity of ABI5 or degrading the ABI5 protein to promote seed germination or postgerminative growth. Conversely, the circadian clock gene *PRR5*[41] and the BR signaling inhibitor *BIN2*[56] inhibited seed germination by promoting the expression of *ABI5* and maintaining its protein stability. Moreover, *ABI5* regulates reactive oxygen species homeostasis by activating *CATALASE1* transcription during seed germination[57]. In the present study, knocking out *ABI5* reduced rice seed dormancy, while overexpressing *ABI5* enhanced rice seed dormancy (Supplementary Fig. 9d-e), indicating that *ABI5* positively regulates rice seed dormancy. SDR3.1 inhibited the transcriptional activity of ABI5 (Fig. 2f-g), thus blocking the signal transduction of ABA, which was not conducive to seed dormancy, and knocking out *SDR3.1* enhanced dormancy.

Interestingly, gene expression experiments showed that *SDR3.1* regulated more ABA signaling genes than did *ABI5* (Fig. 3c-e), suggesting that *SDR3.1* may not only affect seed dormancy through *ABI5*. *ABI3* is another important transcription factor that regulates ABA signaling to affect seed dormancy[36], and the inhibition of ABI3 transcriptional activity by SDR3.1 was confirmed in this study (Supplementary Fig. 13a-b). Therefore, SDR3.1 negatively regulates

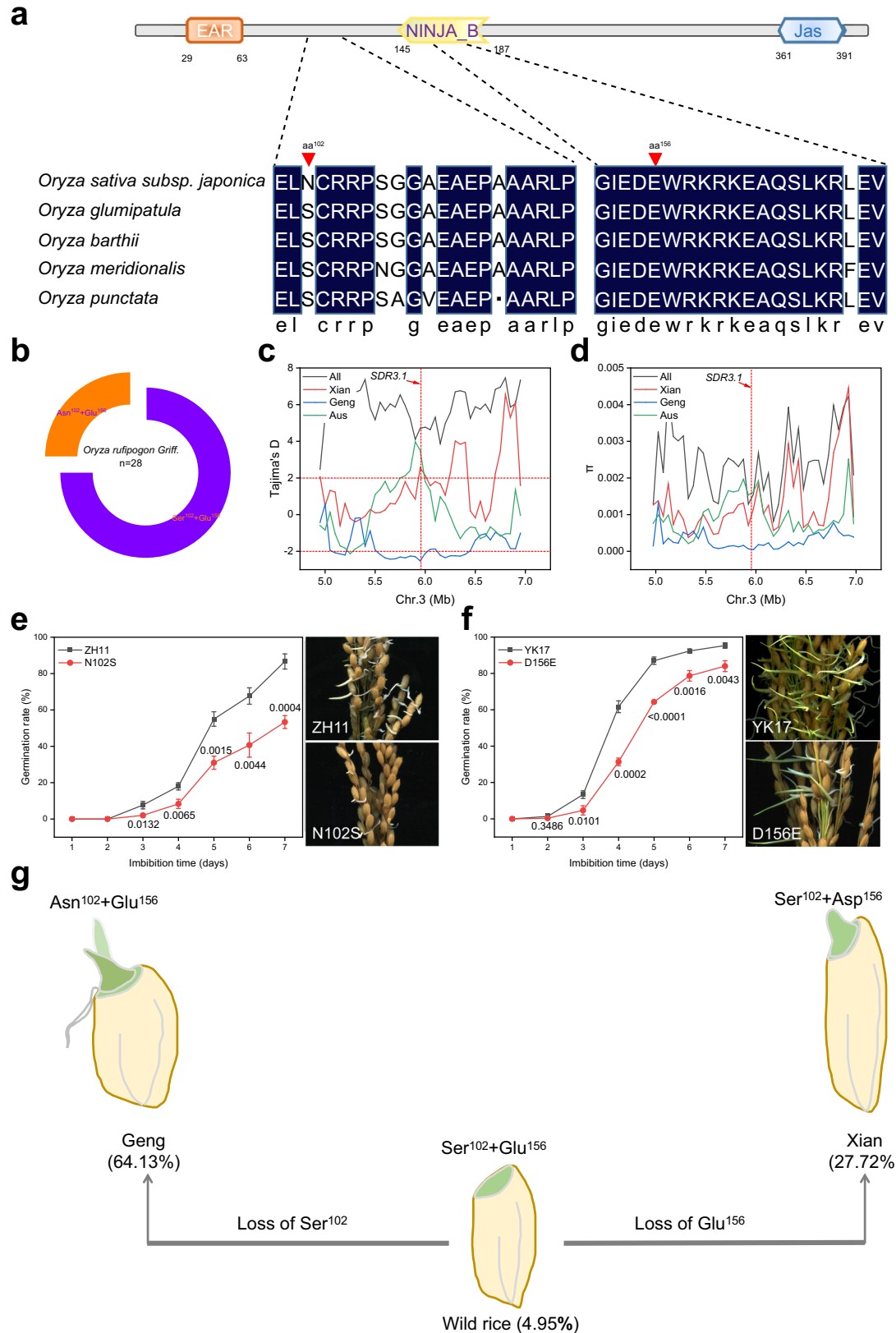

**Fig. 5 | Domestication analysis of *SDR3.1*. a** Analysis of two key amino acids (aa) of SDR3.1 in wild rice. The red arrows mark the positions of the two key amino acids (aa[102] and aa[156]). **b** Two combinations of two key aa of SDR3.1 in 28 wild rice accessions. **c-d** Tajima's D and nucleotide diversity (π) for the -2-Mb genomic region flanking *SDR3.1* in the 3 K RGs; SDR3.1 is indicated by a red dashed line. **e** Germination rate of ZH11 and N102S 35 DAH (*n* = 3 independent experiments); PHS phenotypes of ZH11 and N102S 35 DAH. **f** Germination rate of YK17 and D156E 35 DAH (*n* = 3 independent experiments);

PHS phenotypes of YK17 and D156E 35 DAH. **g** Proposed two-line model for the process of rice seed dormancy domestication. During the domestication of wild rice to *Geng/ japonica*, Ser[102] of *SDR3.1* was converted to Asn[102], which caused most of the dormancy lost. During the process of domestication to *Xian/indica*, Glu[156] was converted to Asp[156], which caused some of the dormancy lost. The values in parentheses indicate the average germination rate. In **e, f** Data are presented as the mean ± SD, and *P* values are indicated by two-tailed Student's *t* test. Source data are provided as a Source Data file.

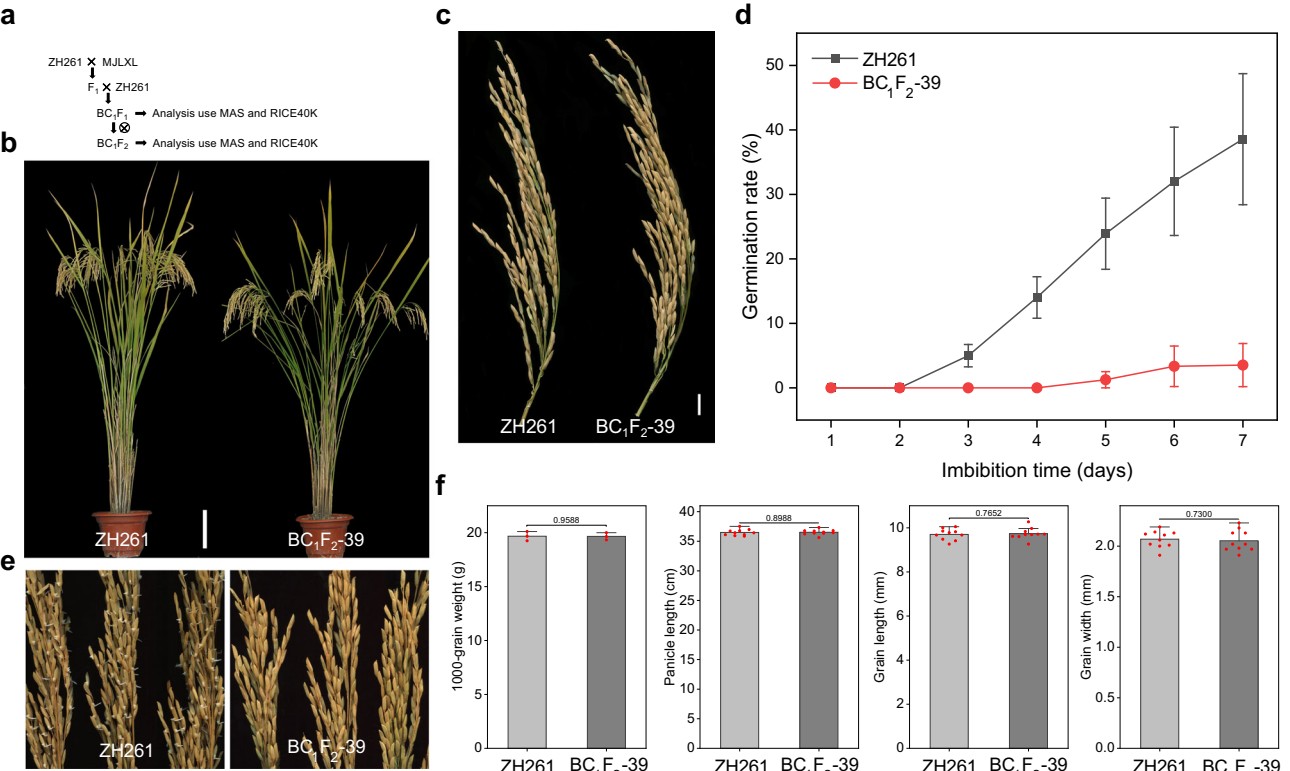

**Fig. 6 | Dormancy improvement of the high-quality rice restorer line. a** The breeding scheme used for the development of BC₁F₂-39 by the introgression of MJLXL (donor parent, DP) into ZH261 (recurrent parent, RP). **b** Plant architectures of ZH261 and BC₁F₂-39 at the grain-filling stage. Bar = 10 cm. **c** Panicle architectures of ZH261 and BC₁F₂-39 at the grain-filling stage. Bar = 5 cm. **d** Germination rate of ZH261 and BC₁F₂-39. **e** PHS phenotypes of ZH261 and BC₁F₂-39. **f** Comparison of agronomic traits between ZH261 and BC₁F₂-39, included the 1000-grain weight, panicle length, grain length, and grain width. Data are presented as the mean ± SD, *P* values are indicated by two-tailed Student's *t* test. *n* = 3 independent experiments (the figure on the left in **f**) or *n* = 10 independent experiments (the three figures on the right in **f**). Source data are provided as a Source Data file.

seed dormancy by inhibiting the transcriptional activity of *ABIs* (Supplementary Fig. 26). In addition, the ABA content in MJLXL was significantly higher than that in ZH11 (Fig. 3f), indicating that the seed dormancy of MJLXL is influenced not only by ABA signaling but also by ABA content. This partly explains why MJLXL seeds hardly germinated on the 7th day (Supplementary Fig. 1a-b), but the germination rate of KO-1 and KO-1 plants reached 10-20% (Supplementary Fig. 4d-e). The seed dormancy mechanism of MJLXL and the molecular mechanism through which *SDR3.1* regulates seed dormancy involve a more complex network, which requires further in-depth research.

Domestication is a process of natural or artificial selection. After 10,000 years of selection, crops such as rice, wheat, corn, soybean, and sunflower have been successfully brought to peoples' tables[58]. Identification of domestication-related genes helps us understand the process of domestication. Nucleotide polymorphism analysis revealed that the π values of *An-1* and *OsNCED3* were reduced in cultivated rice, suggesting that these two genes were subject to artificial domestication[8,49]. In the present study, the π value for *SDR3.1* was reduced, suggesting that *SDR3.1* was subject to artificial domestication. Furthermore, the π value of *Geng/japonica* was the lowest, followed by that of *Xian/indica*, and the π value of *Aus* was the highest (Fig. 5d), suggesting that *SDR3.1* was subjected to stronger selection in *Geng/japonica*, which lost most of its seed dormancy.

The search for functional nucleotide polymorphisms (FNPs) became increasingly relevant as single-base substitution technologies matured[59]. *PAOS*⁻⁵⁷⁸ᴳ, an FNP in the promoter region, reduced the expression level and promoted mesocotyl elongation, which was beneficial for direct seeding of rice[60]. In the present study, haplotype analysis and natural population germination experiments showed that

two FNPs caused two amino acid changes in SDR3.1 protein that had a large effect on seed dormancy (Fig. 4e). Asn¹⁰² and Glu¹⁵⁶ in *Geng/japonica* had the shorter seed dormancy, Ser¹⁰² and Asp¹⁵⁶ in *Xian/indica* had the second longest seed dormancy, and Ser¹⁰² and Glu¹⁵⁶ in *Aus* had the longest seed dormancy (Fig. 4e). Replacing 102nd amino acid in the ZH11 or 156th amino acid in the YK17 significantly reduced the germination rate (Fig. 5e-f), further indicating the importance of aa¹⁰² and aa¹⁵⁶ for seed dormancy.

Homology analysis revealed that *SDR3.1* had the highest homology with wild rice, and most wild rice was a combination of Ser¹⁰² and Glu¹⁵⁶ (Fig. 5a-b). Although molecular phylogenetic analyses suggest that *Xian/indica* and *Geng/japonica* originated independently[61,62], studies of domestication genes suggest a single domestication origin[11,12,14,15,43]. Therefore, the domestication of wild rice into *Geng/japonica* or *Xian/indica* mainly depended on changes in two amino acids: aa¹⁰² was determined for *Geng/japonica* and aa¹⁵⁶ was determined for *Xian/indica* (Fig. 5g). In addition, *Geng/japonica* was first domesticated from wild rice and subsequently hybridized with wild rice from different regions to produce *Xian/indica*[61,63], which explains why a portion of Asn¹⁰² and Glu¹⁵⁶ are combined in wild rice (Fig. 5b).

Domestication is also a process that has led to a dramatic reduction in biodiversity. In the context of the homogenization of the global food supply[64], biodiversity reduction is incompatible with the need for increased crop diversity in the second wave of the Green Revolution; low crop diversity is a threat to food security[65]. The creation of germplasm, including via de novo domestication[66,67], to increase crop diversity is important for food security[68,69]. Therefore, it is essential to retrieve genes lost during domestication and assess their regulatory mechanisms for creating germplasms. In this study, we introduced the

*SDR3.1* allele of MJLXL into ZH261, which was consistent with expectations and significantly reduced the PHS of ZH261 without altering other agronomic traits (Fig. 6). This finding indicates that we successfully created rice germplasm resistant to PHS, which will provide important insights into the problem of PHS in rice planting area in southern China. In addition, pyramiding different seed dormancy genes is a strategy for cultivating rice varieties with resistance to PHS[44]. Our research provides a basis for this.

## Methods

### Plant material and growth conditions

Zhonghua 11 (ZH11), a *Geng/japonica* cultivar, was used as an acceptor, and MJLXL, an *Aus* cultivar from Bangladesh, was used as a donor. The single plant whose genotypes of all markers (108 pairs of SSR polymorphic markers evenly distributed on 12 chromosomes were used for background screening), except markers P1 and P2, were consistent with those of ZH11 was selected as the NIL (from the BC$_4$F$_2$ population). The transgenic plants were all based on ZH11 or YK17. A population composed of 291 rice accessions was used for domestication analysis. The plants were grown in Hangzhou and Hainan under normal field management conditions.

### Evaluation of seed germination

Seed germination was conducted as previously described by He et al.[70]. Briefly, approximately 100 seeds were placed in a 90-mm petri dish covered with two filter papers. Ten milliliters of distilled water was added, and the dishes were placed in a dark incubator at 25 °C for 7 days to allow for germination. Germination was considered to have occurred when the shoot length was half of the seed length or the radicle length was the full length of the seed. Three biological replicates were performed.

### Map-based cloning

Primary gene mapping was conducted in a B$_3$C$_2$ population with 153 plants derived from the cross between ZH11 and MJLXL. Approximately 4146 plants of the BC$_4$F$_2$ population were subjected to recombinant screening with markers P3 and P4. *qSDR3.1* was ultimately narrowed to a 26.9 kb interval between markers Indel27 and Indel29. For the complementation experiments, the entire DNA sequence of *SDR3.1* from ZH11 was inserted into the pCAMBIA1300 vector and subsequently transformed into NIL. The CRISPR/Cas9 vector BGK03 Biogle (Hangzhou, China) was used for genome editing. The primers used in this assay are listed in Supplementary Data 2.

### Expression analysis

Total RNA was extracted with TRIzol (TransGen). RNA reverse transcription was performed using ReverTra Ace (TOYOBO Biotech). qRT–PCR was performed using Applied Biosystems QuantStudio 3 Real-Time PCR System (Thermo Fisher Scientific, USA). The primers used in this assay are listed in Supplementary Data 2.

### Copy number analysis

The copy number of the COMs were detected via qRT–PCR. The pUC plasmid containing the *SDR3.1* gene fragment was used as a standard to generate a standard curve. The primers used in this assay are listed in Supplementary Data 2.

### ABA and GA concentration measurements

A triple quadrupole mass spectrometer coupled with an electrospray ionization and high-performance liquid chromatography (HPLC–ESI–MS/MS) system (Shimadzu LC system, pump model: LC-10ADvp; oven model: CTO-10Avp; system controller model: SCL-10Avp; ABI 4000; Applied Biosystems)[71] was used to quantify phytohormones at Nanjing Ruiyuan Biotechnology Co., Ltd. (Nanjing, China).

### Subcellular localization

The coding sequences of *SDR3.1* without a stop codon were amplified from the cDNA of ZH11, subsequently inserted into pYBA1132. The recombinant *35S::SDR3.1-GFP* vector and the control construct were subsequently transformed into rice protoplasts through polyethylene glycol (PEG4000)-mediated transformation[72]. In brief, rice stem tissues from 11-day-old plants were collected and used as source materials for protoplast extraction; the materials were isolated with an enzymatic digestion solution (0.6 M mannitol, 1.6% cellulose R-10, 0.75% Macerozyme R-10, 0.1% BSA, 10 mM 2-(Nmorpholino)ethanesulfonic acid, 1 mM CaCl$_2$, and 0.4% b-mercaptoethanol). The protoplasts were washed with W5 solution (154 mM NaCl, 125 mM CaCl$_2$·2H$_2$O, 5 mM KCl, 5 mM glucose, 2 mM MES). Plasmid DNA was mixed with the rice protoplasts, and an equal volume of 40% PEG solution (40% [w/v] PEG 4000, 0.1 M CaCl$_2$·2H$_2$O, 0.2 M mannitol) was added and mixed gently. After 20 hours of cultivation under dark conditions at 28 °C, fluorescence was detected with a Zeiss LSM 710 NLO confocal microscope.

### Y2H assay

The coding sequences of *SDR3.1* were inserted into pGBKT7 (Clontech, Dalian, China) as a 'bait' vector, and the coding sequences of *ABI5* were inserted into pGADT7 (Clontech) as a 'prey' vector. The two plasmid vectors were cotransformed into the yeast strain Y2HGold. Transformants were spotted on SD-Leu-Trp media. After 28 °C cultivation for 3 days, five colonies were randomly selected, spotted on SD-Leu-Trp-His-Ade media and incubated for 3 days. The primers used for cloning are listed in Supplementary Data 2.

### Pull-down assay

The coding sequences of *SDR3.1* were inserted into the pGEX-4T-1 vector to create a fusion protein with GST. The coding sequences of *ABI5* were inserted into pET28a to generate a His fusion protein. Both the empty pGEX-4T-1 vector and the corresponding vector were introduced into the *E. coli* strain Rosetta. The GST fusion protein was induced with 0.1 mmol/L isopropyl β-D-1-thiogalactopyranoside (IPTG) at 24 °C for 12 h. The His fusion protein was induced with 1.0 mmol/L IPTG at 16 °C for 12 h. For pull-down assays, 2 μg of His::ABI5 prey protein was incubated with 2 μg of immobilized GST or GST::SDR3.1 bait proteins. The mixtures were incubated on a shaker for 5 h (all operations were performed at 4 °C), after which the proteins were separated via 8% SDS–PAGE. Then the pulled down proteins were immunoblotted with anti-GST or anti-His antibodies. The mouse monoclonal anti-GST, mouse monoclonal anti-His and HRP-goat anti-mouse antibodies used for this study were obtained from Beyotime at a dilution of 1:10,000 (Beyotime: AF0174, AF2870 and A0192). The primers used in this assay are listed in Supplementary Data 2.

### Bimolecular fluorescence complementation assay

The coding sequences of *SDR3.1* were connected to the vector pSPYCE, and the coding sequences of *ABI5* was connected to the vector pSPYNE. The plasmids were subsequently introduced into rice protoplasts[72]. Fluorescence was detected with a Zeiss LSM 710 NLO confocal microscope. The primers used in this assay are listed in Supplementary Data 2.

### Protein extraction and immunoblotting

Total plant protein was extracted from 7-day-old embryos. Extraction buffer (25 mM Tris-HCl, pH=7.4; 150 mM NaCl; 1 mM EDTA; 1% Nonidet P-40; 5% Glycetol; 0.144 g/L protease inhibitor; Roche: 5892791001) (750 μL) was added. The material was dispersed, mixed well, and then centrifuged at 13,000 × g for 20 min; the supernatant was collected. Denatured proteins were separated via SDS–PAGE and transferred to PVDF membranes for immunoblotting with anti-*SDR3.1*. The rabbit

monoclonal anti-*SDR3.1* antibody used for this study was obtained from AtaGenix at a dilution of 1:10,000. The β-actin protein was used as a control. The mouse monoclonal anti-actin, HRP-goat anti-mouse and HRP-goat anti-rabbit antibodies used for this study were obtained from Beyotime and used at a dilution of 1:10,000 (Beyotime: AF0003, A0192 and A0208).

### Y1H assay

The coding sequences of *ABI5* were inserted into pB42AD (Clontech), and the promoter of *EM1* was inserted into pLacz (Clontech). The vectors were subsequently cotransformed into the yeast strain EGY48. Transformants were spotted on SD-Ura-Trp media. After 28 °C cultivation for 3 days, colonies were randomly selected and spotted on SD-Ura-Trp media supplemented with 2% (w/v) galactose, 1% (w/v) raffinose, 1 × salt buffer (7 g/L $Na_2HPO_4 \cdot PH_2O$, 3 g/L $Na_2HPO_4$, pH 7.0), and 80 mg/L 5-bromo-4-chloro-3-indolyl-β-D-galactopyranoside acid (Clontech). The plates were incubated for 1 day. The primers used in this assay are listed in Supplementary Data 2.

### Electromobility shift assay

An electromobility shift assay was performed with a Chemiluminescent EMSA Kit (Beyotime, Shanghai). 2 μg of His::ABI5 protein and 1 μL of biotin-labeled probe (1 ng) were mixed at 25 °C for 20 min. The unlabeled probe (1 ng) was added as a competitor. The sequences of these probes are listed in Supplementary Data 2.

### Transient transactivation assay

The promoter region of *EM1* was inserted into the pGreenII 0800-LUC vector as a reporter plasmid. *ABI5*, *SDR3.1*, and *GFP* were independently inserted into the pGreenII 62-SK vector as effector plasmids. The combined plasmids were introduced into rice protoplasts. The transformed cells were incubated for 24 hours at 28 °C with or without 5 μM ABA, and the relative LUC activity was measured with a dual-luciferase reporter assay instrument (Promega). The primers used for cloning are listed in Supplementary Data 2.

### Evolutionary analysis and population genetic analysis of *SDR3.1*

We used the UniProt database (https://www.uniprot.org/) to obtain proteins homologous to SDR3.1. Species with protein homology greater than 50% were selected for the construction of a phylogenetic tree and comparison with DNAMAN software (Lynnon Biosoft, USA). Haplotype analysis was performed on the 3 K website to determine the haplotype classification of *SDR3.1*, and the coding sequences were chosen as the gene region for analysis. Filtering was performed based on MAF (minor allele frequency) being greater than or equal to 0.01. Cygwin and vcftools (v0.1.16) were used to calculate nucleotide diversity (π) and Tajima's D for the 2-Mb region flanking *SDR3.1* (each 50-kb across the genome as a window and with an overlapping 5-kb step size). A world map was downloaded from Origin 2022 (https://www.originlab.com).

### Cut&Tag sequencing and analysis

We harvested the cells and then centrifuged them for 3 min at 600 × g and 25 °C. Subsequently, the cell nucleus was extracted via CelLytic TM PN Isolation/Extraction Kit (Weibo: 3424255; refer to the manual for the specific steps). Then, the samples were incubated with primary antibodies (mouse anti-GFP-Tag mAb, ABclonal, AE012; dilution 1:100) overnight at 4 °C on a rotating platform. After removing the primary antibody, the cells were incubated at 25 °C for 30 minutes in secondary antibody (HRP-conjugated mouse anti-GFP-Tag mAb, ABclonal, AE030; dilution 1:100) diluted in DigWash buffer. After incubating in pA-Tn5 adapter complex diluent for 1 hour, the nucleus was resuspended in labeled buffer and incubated at 37 °C for 1 hour. The labeling reaction was stopped, and the products were amplified and sequenced with an Illumina NovaSeq 6000, after which the raw data were

obtained. The raw data were evaluated using FastQC software (v0.11.5). Then, Trimmomatic software (v0.39) was used for data filtering, and the adapter sequence at the end was truncated. Next, bwa software (v0.7.17-r1188) was used to perform unique comparisons, screening and duplication processing on the obtained reads. Finally, MACS2 software (v2.2.7.1) was used to find the protein binding sites (peaks).

### Statistical analysis

IBM SPSS Statistics software (v25) was used for Student's *t* test.

### Reporting summary

Further information on research design is available in the Nature Portfolio Reporting Summary linked to this article.

## Data availability

The sequence data from this study can be found at Rice Genome Annotation Project website (http://rice.uga.edu/) under the following accession numbers: *SDR3.1/MODD* (*LOC_Os03g11550* [http://rice.uga.edu/cgi-bin/ORF_infopage.cgi?orf=LOC_Os03g11550]), *OsABI5* (*LOC_Os01g64000* [http://rice.uga.edu/cgi-bin/ORF_infopage.cgi?orf=LOC_Os01g64000]), *OsABI3* (*LOC_Os01g68370* [http://rice.uga.edu/cgi-bin/ORF_infopage.cgi?orf=LOC_Os01g68370]), *OsEM1* (*LOC_Os05g28210* [http://rice.uga.edu/cgi-bin/ORF_infopage.cgi?orf=LOC_Os05g28210]), *OsLEA3* (*LOC_Os05g46480* [http://rice.uga.edu/cgi-bin/ORF_infopage.cgi?orf=LOC_Os05g46480]), *OsTRAB1* (*LOC_Os08g36790* [http://rice.uga.edu/cgi-bin/ORF_infopage.cgi?orf=LOC_Os08g36790]), *OsRab16A* (*LOC_Os11g26790* [http://rice.uga.edu/cgi-bin/ORF_infopage.cgi?orf=LOC_Os11g26790]), and *OsbZIP72* (*LOC_Os09g28310* [http://rice.uga.edu/cgi-bin/ORF_infopage.cgi?orf=LOC_Os09g28310]). Source data are provided with this paper.

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

## Acknowledgements
We thank Ling Jiang (NanJing Agricultural University), Zhigang Zhao (NanJing Agricultural University) and Jiuyou Tang (Institute of Genetics and Developmental Biology, Chinese Academy of Sciences) for their valuable suggestions and revisions to the manuscript. This research was financially supported by the National Natural Science Foundation of China (No.32188102, 31871597, 32071991), the Zhejiang Science and Technology Major Program on Agricultural New Variety Breeding (2021C02063-2), the Key Research and Development Program of the China National Rice Research Institute (CNRRI-2020-02), Zhejiang Provincial Natural Science Foundation of China (LDQ23C130001), Zhejiang Provincial Science and Technology Project (2020R51007), the Key Research and Development Program of Zhejiang Province (2022C02011).

## Author contributions
H.P. and S.Z. designed the research; G.N., T.S., W.Y., C.W., A.R., R.Z., H.S., T.S., W.X., S.G., J.G., X.L., W.L., C.Y. and Z.F. performed the experiments; S.Z. and G.N. analyzed the results and wrote the manuscript.

## Competing interests
The authors declare no competing interests.
