## [Peer Review File · Nature Communications]

A mediator of OsbZIP46 deactivation and degradation negatively regulates seed dormancy in riceReviewers' Comments:

Reviewer #1:

Remarks to the Author:

The authors report the detection of qSDR3.1, a quantitative trait locus for seed dormancy in rice. They found that qSDR3.1 interacts with ABI5 and represses seed dormancy by inhibiting the transcriptional activity of ABI5. Natural variation analysis reveals that qSDR3.1 was selected during domestication. However, the genetic analysis of qSDR3.1 and its interaction with ABI5 is not robust. The effect of two SNPs that caused two amino acid changes in qSDR3.1 on dormancy needs additional evidence of experimental data. Thus, this version of the manuscript is not acceptable for publication. Below are some issues the authors would consider when revising the manuscript.

1. The title "Novel quantitative trait locus qSDR3.1..." is not appropriate because qSDR3.1 is the gene MODD that was reported previously.
2. Line 83-84, the term "germination" should be "germination rate".
3. Line 88, "seed vigor" should be "seed viability".
4. Fig.1b and Line 110, the interval of 529 kb is too large to deduce MODD as the candidate of SDR3.1. This deduction needs more evidence.
5. Line 98, the statement of a recessive allele from MJLXL is inaccurate, given the results in Fig. s3b, of which SDR3.1 is an incomplete dominant gene.
6. Line 117, need to describe how to obtain the "six chromosome segment substitution lines".
7. in Line 125, there is a large difference in germination rate (90.1% vs. 50.9%) between the two complementary lines. The authors should explain or discuss that.
8. Line 123-125, why not introduce the MJLXL allele into ZH11 for the complementation test?
9. Line 162, please provide some background of MODD (e.g., reported by Tang et al. 2016) in the Introduction section. Otherwise, it is difficult to follow the logical way to determine the interaction between SDR3.1 and ABI5, because there are many genes involved in the ABA pathway that regulate seed dormancy.
10. Line 197, A similar suggestion for EM1. Why the author used only the gene EM1 to analyze the biological function of SDR3.1 interacting with ABI5. The other biosynthetic or metabolic genes in the ABA pathway could be selected as the marker genes.
11. Fig 3. please clarify what short horizontal lines mean in the histogram of frequency distribution.
12. P289-303, the function of aa102 and aa156 of sdr3 needs supporting data from experiments
13. Please provide notes for the number along each chromosome in Fig. S2.
14. P263-264, 300-301, the sentences are incomplete.
15. P305-318, Fig. S21, as the graphical genotype shows BC1F2-39 line carried several chromosomal segments from MJLXL, it needs validation experiments to support that only qSDR3.1 increased seed dormancy in the improved line.

Reviewer #2:

Remarks to the Author:

Comments on the manuscript entitled "Novel quantitative trait locus qSDR3.1 represses *Oryza sativa*. L seed dormancy" by Guo et al.

General comments:

Domestication of cereal crops eliminated majority of seed dormancy and thus research on natural variation for the domestication-related trait in crop species could provide information on the underlying genes and regulatory mechanisms to solve germination-related problems, such as vivipary, pre-harvest sprouting and inadequate germination, in crop production. The manuscript reported three quantitative trait loci (QTL) associated with seed dormancy in a segregating population from a cross between an Aus-ecotype and a japonica-type variety of rice. One of the QTL was delimited to a genomic region of ~500 kb containing >50 predicted genes, including the previously reported Mediator of OsbZIP46 Deactivation and Degradation (MODD) involved in the signaling of abscisic acid,

a dormancy-inducing hormone. Thus, a series of experiments were conducted to confirm the gene's function, to characterize the gene or gene's product for cellular localization, interaction with the ABA signaling gene ABI5, and haplotypic variation in rice germplasm. This research discovered the MODD locus as a seed dormancy QTL and provided lines of information to infer a regulatory role of the QTL underlying gene in the trait development and allelic distribution of the gene in the crop species. Seed dormancy is a complex trait, which requires a reasonable size of plant samples/populations to test for linkage disequilibrium between a QTL and closely linked markers and to estimate quantitative genetic parameters for individual lines or recombinant genotypes. In addition, a QTL consists of functionally differentiated alleles. Thus, the research could be focused on functional site mutation(s) at the QTL underlying gene. Finally, the manuscript requires extensive revision for some concepts, the English language (including the title), and data annotation. naturally

Specific comments/suggestions:

1. Introduction: Add a summary on MODD to indicate what are known and unknown about the gene and refine the research objectives accordingly.
2. Lines 81-89: 1) indicate if n was the number of plants or bulked seed samples in Fig. S1; and 2) the sample size of n=3 is too small to assess the genotypic difference in seed dormancy by germination testing in this and the following experiments.
3. Lines 90-99: 1) add genotyping and phenotypic (frequency distribution) data for the mapping population of 153 BC3F3(?) lines to map the three QTLs; 2) add the genotyping data for the BC3F1 plant(?) used to develop the mapping population, which is critical to annotate the data in Fig. 1b; 3) indicate if the linkage map was constructed based on the mapping population; 4) list parameters for the QTLs in a table; 5) the map resolution for chromosome 3 can be improved using the markers on Fig. 1b; and 6) add likelihood distributions for the other chromosomes or chromosomal segments segregating in the population and move the distributions to Fig. 1, as this information is more important than the pedigree in Fig. 1 in this research.
4. Lines 100-107: 1) indicate if the 4146 BC3F3 plants were genotyped for all the markers; if genotyped, >8000 gametes can be used to map the QTL in a resolution of about 1/8000 Morgan, or <0.1 cM, or <25 Kb (not >500 kb), or a few (not >50) candidate genes; and 2) indicate in the legends to Fig. 1b if the Rs are single plants or pure lines; if they were lines, indicate the number of plants used to evaluate the means and sd. It is not common for two contrast genotypes at a QTL to produce the difference of ~70% (Fig. 1b) in the same environment. These estimates may include effects of the backgrounds or other factors.
5. Lines 100-116: add cDNA sequences from the parents to confirm the gene model.
6. Lines 117-114: 1) explain why the complementation test was conducted using this allele, not the other allele, or both; 2) add information on the transgenic lines, such as the generation and copy numbers; and 3) add a control to the test.
7. Lines 129-136: 1) add information on how the two mutants were selected; and 2) indicate if the protein sequence was deduced based on the genomic DNA sequence and the gene model of the reference, or based on the cDNA sequences, and if the mutant lines were sequenced for the entire gene or for the mutant-containing fragments only.
8. Lines 137-144: Add statistic data to quantify the genotypic differences. Data from a controlled experiment can help support the point.
9. Lines 146-151: Indicate what lines were used for the analysis and if there were data from germinating seeds.
10. For molecular biological experiments, add information on the experimental procedure/conditions (GFP expression, pull-down assay, protein extraction and immunoblotting) and cite a reference for methods (BiFC).

Reviewer #3:

Remarks to the Author:

The manuscript by Guo et al. reported the identification of QTL that modulates the seed dormancy in rice. Firstly, the authors identified the QTL qSDR3.1 by positional cloning and transgenic approaches.

They also found that qSDR3.1 negatively regulated seed dormancy by reducing the transcriptional activity of ABI5. They also tested whether the qSDR3.1 was targeted during rice domestication. Finally, the authors evaluated the potential of qSDR3.1 in rice breeding application. It is interesting to find the novel functions of qSDR3.1/MODD for seed dormancy in rice. However I still have several concerns as below.

Major comments:

Page 5, Line 98

The authors considered the qSDR3.1 allele from MJLXL is recessive based on the genetic analysis of BC3F2 family. However, Supplementary Fig. 3b did not support this inference.

Page 5, Line 102

The authors obtained six recombinants between markers P3 and P4 from 4146 BC4F2 individuals and fine mapped qSDR3.1 to a 529 kb interval. Please provide some evidence to demonstrate that the chromosome region harboring qSDR3.1 has lower recombinant ratio compared with rice whole genome.

Figure 1b and c

The authors performed phenotype comparison of paired lines. Please explain why the two lines were paired. For example, R1 and R2, R3 and R4.

Please provide the genotypes or nucleotide sequences of LOC_Os03g11550 of the three recombinants, including R2, R3, and R4, and confirm that the genotypes of LOC_Os03g11550 are cosegregated with the phenotype of seed dormancy.

Page 12, line 241

Asian cultivated rice have a dramatic differentiation of indica and japonica. So, the evaluation of genetic effect of different alleles seriously affected by population structure. I suggest that the authors might compare the seed dormancy in the indica and japonica varieties in both Hap1 and Hap2 with Ser102+Asp156, and judge the influence of population structure on the analysis of genetic factor for seed dormancy using rice natural population. In addition, the authors might investigate the seed dormancy of 28 accessions of wild rice (Fig. 4b), and compare the phenotype in the Asn102+Glu156 and Ser102+Glu156 groups.

Page 15 line 298

It is suggested that the authors analyze whether qSDR3.1 is selected during rice domestication, and then analyze the selection characteristics of chromosomal region carrying qSDR3.1 gene.

EM1 is an important downstream gene of the regulatory pathway of seed dormancy described in this paper. So, the authors need investigate the expression level of EM1 in genetic materials and evaluate the seed dormancy of transgenic plants, including knock out and overexpression.

The findings in this study indicated that the qSDR3.1 interacts with ABI5, regulating ABA signaling, thereby modulating seed dormancy in rice. So, the authors might investigate and compare the expression of ABA response genes and ABA content in ZH11, MJLXL, and transgenic plants.

Minor comments:

In the title, "Oryza sativa. L" should be "Oryza sativa L.". "Novel" should be removed.

Page 6, line 124 and Figure 1d

The germination rate showed an obvious difference in the complementary transgenic lines COM-1 (90%) and COM-2 (50%). Please investigate the seed dormancy of more than three Independent transgenic lines and explain the reason for the phenotypic difference of transgenic lines harboring the same construct.

Page 11, line 212

The authors found that the knockout mutants of qSDR3.1 showed a decrease of ABI5 protein content compared with the wild type. Please explain why the mutants with low ABI5 protein content had a strong transcription activity.

Page 11, line 228

The authors did not describe the results of haplotype analysis in this paragraph.

Page 14, line 292

The results of amino acid sequences analysis of wild rice are not included in this Supplementary Fig. 11.

Page 19 line 405

"embryo" should be "shoot".

Fig. 1e

Please describe the genotype of this NIL.

Figure S6

The germination rate of ZH11 had an obvious difference in panel a and b.

Figure S7b

It is difficult to judge that the qSDR3.1-GFP signal was localized in the nucleus from these pictures. The nucleus-localized marker should be used in the subcellular location experiment.

Reviewer #4:

Remarks to the Author:

The results in MS "Novel quantitative trait locus qSDR3.1 represses *Oryza sativa*. L seed dormancy" shown that discovery of qSDR3.1, a new 18 quantitative trait locus (QTL) for seed dormancy in rice. knockout lines of qSDR3.1 exhibited significant reduction in PHS. The qSDR3.1 locus negatively regulated seed dormancy by directly inhibiting the transcriptional activity of ABSCISIC ACID INSENSITIVE5 (ABI5). All data are interesting. Although the qSDR3.1 loci, LOC_Os03g11550, annotated as Mediator of OsbZIP46 Deactivation and Degradation(MODD), was reported which mediates the inactivation and degradation of OsbZIP46 and negatively regulates ABA signaling and drought resistance in rice, but the authors found its new function in seed dormancy.

Major question:

1. In Figure S1 and Figure S4, the seeds of MJLXL were not germinated at 7d, but the germinating rate of knockout mutant seeds reached 5%-10%. please discuss the results.
2. The authors thought that the interaction between qSDR3.1 and ABI5 to regulate seed dormancy. In general, the ABI5 protein in Arabidopsis was not involved in seed dormancy. Please show the data and phenotype of rice ABI5 mutant.
3. To verify the relation of between qSDR3.1 and ABI5, please construct these double mutants and analysis these mutant phenotype.
4. Figure S9 shown that the ABI5 proteins binded with the promoter of EM1, but in many research works the ABI5 was also binded with the promoter of EM6. Please check whether ABI5 is able to bind the promoter of EM6 to be involved in seed dormancy? and please check other genes involved in seed dormancy.
5. In Introduction section, the authors said that seed dormancy was regulated by ABA and GA signaling, please check the ABA and GA contents of wild-type and mutants?

Dear reviewers

Thank you very much for your time and comments on our manuscript. Those comments are all valuable and very helpful for getting the most out of our work. Our manuscript “Novel quantitative trait locus *qSDR3.1* represses *Oryza sativa*. L seed dormancy” Manuscript ID: NCOMMS-22-48784B, has been revised carefully and marked it in red and here is the listed response to the reviewers’ comments.

Thanks again for your time on our manuscript and any further comments from you and the four reviewers will be highly appreciated!

With best regards,

Sincerely yours,

Corresponding author: Zhonghua Sheng and Peisong Hu

First author: Naihui Guo

Response to reviewers:

Reviewer #1 (Remarks to the Author):

The authors report the detection of *qSDR3.1*, a quantitative trait locus for seed dormancy in rice. They found that *qSDR3.1* interacts with *ABI5* and represses seed dormancy by inhibiting the transcriptional activity of *ABI5*. Natural variation analysis reveals that *qSDR3.1* was selected during domestication. However, the genetic analysis of *qSDR3.1* and its interaction with *ABI5* is not robust. The effect of two SNPs that caused two amino acid changes in *qSDR3.1* on dormancy needs additional evidence of experimental data. Thus, this version of the manuscript is not acceptable for publication. Below are some issues the authors would consider when revising the manuscript.

Response: Thank you very much for the reviewer's comments that important for improving our MS. We have strengthened the interaction relationship between *qSDR3.1* and *ABI5* through additional experiments (reply to point 10), and added additional experiments to confirm the effect of two SNPs on dormancy (reply to point 12). We hope that our new evidence can fully address the concerns of the reviewer.

1.The title "Novel quantitative trait locus *qSDR3.1*..." is not appropriate because *qSDR3.1* is the gene *MODD* that was reported previously.

Response: Thanks for the reviewer’s comments. We have changed the title to "The quantitative trait locus *qSDR3.1* as a brake represses seed dormancy in rice". (page 1, line 1)

2. Line 83-84, the term "germination" should be "germination rate".

Response: Thanks for the reviewer's careful observation. "germination" has been revised into "germination rate". (page 5, line 94-95)

3. Line 88, "seed vigor" should be "seed viability".

Response: Thanks for the reviewer's careful observation. "seed vigor" has been revised into "seed viability". (page 5, line 99)

4. Fig.1b and Line 110, the interval of 529 kb is too large to deduce *MODD* as the candidate of *SDR3.1*. This deduction needs more evidence.

Response: Thanks for the reviewer's comments. We designed 5 pairs of Indel markers with polymorphism between parents within the 529-kb region (Add the original 6 pairs of markers, a total of 11 pairs of markers; page 6, line 115), and ultimately located the gene within 26.9-kb region between Indel27 and Indel29 (Additionally, six homozygous recombinant lines were selected, also named R1-R6). This region contains 4 candidate genes. After gene function prediction and sequencing analysis, we determined the candidate gene with *MODD* as *qSDR3.1*. (page 6, line 120-135; Fig. 1b)

5. Line 98, the statement of a recessive allele from MJLXL is inaccurate, given the results in Fig. s3b, of which *SDR3.1* is an incomplete dominant gene.

Response: Thanks for the reviewer's comments. Yes, *qSDR3.1* is an incomplete dominant gene. We have made revisions in the manuscript. (page 6, line 111-113)

6. Line 117, need to describe how to obtain the "six chromosome segment substitution lines".

Response: Thanks for the reviewer's comments. We selected six chromosome segment substitution lines from the BC₃F₃ population. (page 7, line 136-137)

7. in Line 125, there is a large difference in germination rate (90.1% vs. 50.9%) between the two complementary lines. The authors should explain or discuss that.

Response: Thanks for the reviewer's comments. The copy number results showed that the copy number of COM-1 was almost twice that of COM-2 and COM-3 (Fig. 1e), and copy number variation would affect gene function. Therefore, we speculated that the significant difference in germination rate among COM-1, COM-2 and COM-3 was due to copy number variation. (page 7, line 143-144; Page 19, line 403-408)

8. Line 123-125, why not introduce the MJLXL allele into ZH11 for the complementation test?

Response: Thanks for the reviewer's comments. We used the ZH11 allele to complement NIL, but did not use the MJLXL allele to complement ZH11 because the significant difference in genetic background between MJLXL and ZH11, when the MJLXL allele is introduced into ZH11 alone, its phenotype is susceptible to interference from the genetic background of ZH11, which is not conducive to evaluating the results.

9. Line 162, please provide some background of *MODD* (e.g., reported by Tang et al. 2016) in the Introduction section. Otherwise, it is difficult to follow the logical way to determine the interaction between *SDR3.1* and *ABI5*, because there are many genes involved in the ABA pathway that regulate seed dormancy.

Response: Thanks for the reviewer's comments. We have added some background of *MODD* to the introduction. *MODD* is homologous to the *Arabidopsis thaliana* *ABI5* binding protein AFP, so we speculate that *MODD* also interacts with *ABI5* in rice. (page

4, line 76-79)

10. Line 197, A similar suggestion for *EMI*. Why the author used only the gene *EMI* to analyze the biological function of *SDR3.1* interacting with *ABI5*. The other biosynthetic or metabolic genes in the ABA pathway could be selected as the marker genes.

Response: Thanks for the reviewer's suggestion. Through cut&Tag, Y1H, EMSA, and LUC experiments, we identified *LEA3* as another downstream marker gene of *ABI5* to analyze the relationship between *qSDR3.1* and *ABI5*. (Supplementary Fig. 10d-f; page 11-12, line 238-241).

11. Fig 3. please clarify what short horizontal lines mean in the histogram of frequency distribution.

Response: Thanks for the reviewer's comments. The horizontal lines in Figures 4c, e and g represented the number of varieties and have been added to the annotations in Figure 4. (page 44, line 873-874, 876 and 878-879)

12. P289-303, the function of aa¹⁰² and aa¹⁵⁶ of *SDR3.1* needs supporting data from experiments.

Response: Thanks for the reviewer's comments.

Firstly, we found significant differences in germination rates among different combinations of aa¹⁰² and aa¹⁵⁶. (Fig. 4e)

Secondly, we utilized *EMI* and *LEA3* as downstream marker genes, and found that Hap3 and Hap4 had a stronger inhibitory effect on *ABI5* than Hap5, due to the difference in aa¹⁰². And the inhibitory effect of Hap1 and Hap2 on *ABI5* is also stronger than Hap5, but not as strong as Hap3 and Hap4, which is due to the difference in aa¹⁵⁶. (Supplementary Fig. 24)

Finally, we constructed site-directed-mutation mutant containing the aa102 or aa156 substitutions: that is, we changed the asparagine (N) to the serine (S) at the 102 site of ZH11 and changed the aspartic acid (D) to the glutamic acid (E) at the 156 site of YK17. The results showed that compared with ZH11 or YK17, the germination rates of the mutants was significantly reduced (Fig. 5e-f). Therefore, the function of aa¹⁰² and aa¹⁵⁶ of *SDR3.1* is important for seed dormancy.

13. Please provide notes for the number along each chromosome in Fig. S2.

Response: Thanks for the reviewer's comments. The number on the left of chromosome represents the physical distance (Mb), the number on the right of the chromosome represents the marker name and we annotated them in Figure S2.

14. P263-264, 300-301, the sentences are incomplete.

Response: Thanks for the reviewer's comments. We have completed the sentences in the corresponding positions of the manuscript. (page 15, line 324-325; page 17, line 363-365)

15. P305-318, Fig. S21, as the graphical genotype shows BC₁F₂-39 line carried several chromosomal segments from MJLXL, it needs validation experiments to support that only *qSDR3.1* increased seed dormancy in the improved line.

Response: Thanks for the reviewer's comments. In the background of BC₁F₂-39, it is shown that in addition to the *qSDR3.1* region, the tail of chromosome 3, parts of chromosomes 5 and 7, and the end of chromosome 12 are derived from MJLXL. And

our mapping results show that there are no dormancy QTLs in these regions (Supplementary Fig. 2b), so it is unlikely that these substitution regions have increased the dormancy of the improved lines. In addition, we also investigated two sister lines of BC₁F₂-39, BC₁F₂-27 and BC₁F₂-57, which showed a significant increase in dormancy. However, in their background, the region from MJLXL was different from BC₁F₂-39 except for the region of *qSDR3.1* (Supplementary Fig. 25d-i). In summary, only *qSDR3.1* increased the dormancy of the improved line.

Reviewer #2 (Remarks to the Author):

Comments on the manuscript entitled “Novel quantitative trait locus *qSDR3.1* represses *Oryza sativa*. L seed dormancy” by Guo et al.

General comments:

Domestication of cereal crops eliminated majority of seed dormancy and thus research on natural variation for the domestication-related trait in crop species could provide information on the underlying genes and regulatory mechanisms to solve germination-related problems, such as vivipary, pre-harvest sprouting and inadequate germination, in crop production. The manuscript reported three quantitative trait loci (QTL) associated with seed dormancy in a segregating population from a cross between an Aus-ecotype and a japonica-type variety of rice. One of the QTL was delimited to a genomic region of ~500 kb containing >50 predicted genes, including the previously reported Mediator of OsZIP46 Deactivation and Degradation (MODD) involved in the signaling of abscisic acid, a dormancy-inducing hormone. Thus, a series of experiments were conducted to confirm the gene's function, to characterize the gene or gene's product for cellular localization, interaction with the ABA signaling gene ABI5, and haplotypic variation in rice germplasm. This research discovered the MODD locus as a seed dormancy QTL and provided lines of information to infer a regulatory role of the QTL underlying gene in the trait development and allelic distribution of the gene in the crop species.

Seed dormancy is a complex trait, which requires a reasonable size of plant samples/populations to test for linkage disequilibrium between a QTL and closely linked markers and to estimate quantitative genetic parameters for individual lines or recombinant genotypes. In addition, a QTL consists of functionally differentiated alleles. Thus, the research could be focused on functional site mutation(s) at the QTL underlying gene. Finally, the manuscript requires extensive revision for some concepts, the English language (including the title), and data annotation. Naturally Specific comments/suggestions:

Response: Thank you for reviewer's comments that greatly improve the quality and readability of the MS. I hope our revision can fully address the concerns of the reviewer.

1. Introduction: Add a summary on *MODD* to indicate what are known and unknown about the gene and refine the research objectives accordingly.

Response: Thanks for the reviewer's comments. We have added some background of *MODD* (include what are known and unknown) to the introduction. (page 4, line 76-79)

2. Lines 81-89: 1) indicate if n was the number of plants or bulked seed samples in Fig.

S1; and 2) the sample size of $n=3$ is too small to assess the genotypic difference in seed dormancy by germination testing in this and the following experiments.

Response: Thanks for the reviewer's comments. 1) n means bulked seed samples and $n=3$ independent experiments, refer to "Parallel selection on a Dormancy gene during domestication of crops from multiple families" (NG,2018). 2) As described in the manuscript, we used seeds from the same period for germination experiments, and the maturity of these seeds was consistent. Then, approximately 100 seeds for an independent experiment (materials and methods), and three replicates have approximately 300 seeds, so this sample size can be used to evaluate the experimental results.

3. Lines 90-99: 1) add genotyping and phenotypic (frequency distribution) data for the mapping population of 153 $BC_3F_2(?)$ lines to map the three QTLs; 2) add the genotyping data for the BC_3F_1 plant(?) used to develop the mapping population, which is critical to annotate the data in Fig. 1b; 3) indicate if the linkage map was constructed based on the mapping population; 4) list parameters for the QTLs in a table; 5) the map resolution for chromosome 3 can be improved using the markers on Fig. 1b; and 6) add likelihood distributions for the other chromosomes or chromosomal segments segregating in the population and move the distributions to Fig. 1, as this information is more important than the pedigree in Fig. 1 in this research.

Response: Thanks for the reviewer's comments. 1) We have added genotype and phenotype distribution maps for 153 lines in Supplementary Figure 2a; 2) We have added the genotype of BC_3F_1 in Supplementary Table 3; 3) the linkage map was constructed based on the mapping population; 4) We have added the parameters for the QTLs in Supplementary Table 4; 5) We have used the markers on Figure 1b to improve the map resolution of chromosome 3 (Supplementary Fig. 2b); 6) the BC_3F_2 population was used to mapping dormancy related QTLs, except the target chromosome region, the other chromosome regions were mostly homozygous as recurrent parent, so nearly have not other chromosomes or chromosomal segments segregating in the population.

4. Lines 100-107: 1) indicate if the 4146 BC_3F_3 plants were genotyped for all the markers; if genotyped, >8000 gametes can be used to map the QTL in a resolution of about $1/8000$ Morgan, or <0.1 cM, or <25 Kb (not >500 kb), or a few (not >50) candidate genes; and 2) indicate in the legends to Fig. 1b if the Rs are single plants or pure lines; if they were lines, indicate the number of plants used to evaluate the means and sd. It is not common for two contrast genotypes at a QTL to produce the difference of $\sim 70\%$ (Fig. 1b) in the same environment. These estimates may include effects of the backgrounds or other factors.

Response: Thanks for the reviewer's professional comments. 1) Yes, 4146 individual plants were mainly genotyped for markers between P3 and P4, but the narrowing of the interval was stopped due to the absence of polymorphic markers between the parents between P1 and P2 markers. Now we have resequenced the parents and added 5 pairs of polymorphic Indel markers between P1 and P2 (Add the original 6 pairs of markers, a total of 11 pairs of markers; page 6, line 115), further narrowing the interval of *qSDR3.1* to 26.9-kb (Additionally, six homozygous recombinant lines were selected, also named R1-R6), which contains 4 candidate genes (Fig. 1b). 2) The Rs are single

plants. We are also curious that the difference is close to 70%. Perhaps, as the reviewer said, this result is influenced by background or other factors.

5. Lines 100-116: add cDNA sequences from the parents to confirm the gene model.

Response: Thanks for the reviewer's comments. We added the parental cDNA in Supplementary Fig. 3.

6. Lines 117-114: 1) explain why the complementation test was conducted using this allele, not the other allele, or both; 2) add information on the transgenic lines, such as the generation and copy numbers; and 3) add a control to the test.

Response: Thanks for the reviewer's comments. 1) Because NIL introduced the *qSDR3.1* allele of MJLXL in the genetic background of ZH11, we complementary the *qSDR3.1* allele of ZH11 into NIL; 2) We have added the generation and copy numbers data (page 7, line 143-146); 3) We used ZH11 as a control.

7. Lines 129-136: 1) add information on how the two mutants were selected; and 2) indicate if the protein sequence was deduced based on the genomic DNA sequence and the gene model of the reference, or based on the cDNA sequences, and if the mutant lines were sequenced for the entire gene or for the mutant-containing fragments only.

Response: Thanks for the reviewer's comments. 1) two homozygous mutants KO-1 and KO-2 were screened through sequencing (page 7, line 150-151); 2) The protein sequence is derived based on cDNA sequence, and the mutant lines were sequenced for the mutant-containing fragments.

8. Lines 137-144: Add statistic data to quantify the genotypic differences. Data from a controlled experiment can help support the point.

Response: Thanks for the reviewer's comments. We have added relevant data to Supplementary Figure 6b. (page 8, line 161-164)

9. Lines 146-151: Indicate what lines were used for the analysis and if there were data from germinating seeds.

Response: Thanks for the reviewer's comments. We analyzed these data using ZH11, and the germination data of ZH11 is shown in Supplementary Figure 1.

10. For molecular biological experiments, add information on the experimental procedure/conditions (GFP expression, pull-down assay, protein extraction and immunoblotting) and cite a reference for methods (BiFC).

Response: Thanks for the reviewer's comments. We have added the information on the experimental procedure/conditions (GFP expression, pull-down assay, protein extraction and immunoblotting) and cite a reference for methods (BiFC). (Materials and Methods).

Reviewer #3 (Remarks to the Author):

The manuscript by Guo et al. reported the identification of QTL that modulates the seed dormancy in rice. Firstly, the authors identified the QTL *qSDR3.1* by positional cloning and transgenic approaches. They also found that *qSDR3.1* negatively regulated seed dormancy by reducing the transcriptional activity of ABI5. They also tested whether the *qSDR3.1* was targeted during rice domestication. Finally, the authors evaluated the potential of *qSDR3.1* in rice breeding application. It is interesting to find the novel

functions of *qSDR3.1/MODD* for seed dormancy in rice. However I still have several concerns as below.

Response: Thank you very much for reviewer's positive comments of this MS. We have revised the MS according to the reviewer's comments, hoping to fully address the reviewer's concerns.

Page 5, Line 98

The authors considered the *qSDR3.1* allele from MJLXL is recessive based on the genetic analysis of BC₃F₂ family. However, Supplementary Fig. 3b did not support this inference.

Response: Thanks for the reviewer's comments. Yes, *qSDR3.1* is an incomplete dominant gene. We have made revisions in the manuscript. (page 6, line 112-113)

Page 5, Line 102

The authors obtained six recombinants between markers P3 and P4 from 4146 BC₄F₂ individuals and fine mapped *qSDR3.1* to a 529 kb interval. Please provide some evidence to demonstrate that the chromosome region harboring *qSDR3.1* has lower recombinant ratio compared with rice whole genome.

Response: Thanks for the reviewer's comments. The chromosome region harboring *qSDR3.1* has not lower recombinant ratio compared with rice whole genome. The narrowing of the interval was stopped due to the absence of polymorphic markers between the parents between P1 and P2 markers. Now we have resequenced the parents and added 5 pairs of polymorphic Indel markers between P1 and P2 (Add the original 6 pairs of markers, a total of 11 pairs of markers; page 6, line 115), further narrowing the interval of *qSDR3.1* to 26.9-kb (Additionally, six homozygous recombinant lines were selected, also named R1-R6), which contains 4 candidate genes (Fig. 1b)

Figure 1b and c

The authors performed phenotype comparison of paired lines. Please explain why the two lines were paired. For example, R1 and R2, R3 and R4.

Please provide the genotypes or nucleotide sequences of *LOC_Os03g11550* of the three recombinants, including R2, R3, and R4, and confirm that the genotypes of *LOC_Os03g11550* are cosegregated with the phenotype of seed dormancy.

Response: Thanks for the reviewer's comments. The presentation of two lines in pairs is to demonstrate significant differences in phenotype (germination rate). Refer to "Fine mapping a quantitative trait location, *qSER-7*, that controls stigma exertion rate in rice (*Oryza sativa* L.)"(Rice,2019).

We have added the CDS of *LOC_Os03g11550* of the three recombinants in Supplementary Fig. 3.

Page 12, line 241

Asian cultivated rice have a dramatic differentiation of *Indica* and *Japonica*. So, the evaluation of genetic effect of different alleles seriously affected by population structure. I suggest that the authors might compare the seed dormancy in the indica and japonica varieties in both Hap1 and Hap2 with Ser¹⁰²+Asp¹⁵⁶, and judge the influence of population structure on the analysis of genetic factor for seed dormancy using rice natural population. In addition, the authors might investigate the seed dormancy of 28 accessions of wild rice (Fig. 4b), and compare the phenotype in the Asn¹⁰²+Glu¹⁵⁶ and

Ser¹⁰²+Glu¹⁵⁶ groups.

Response: Thanks for the reviewer's suggestions. We have compared the seed dormancy in the *Indica* and *Japonica* varieties in both Hap1 and Hap2 with Ser¹⁰²+Asp¹⁵⁶, and the results showed that the germination rate of *Japonica* was significantly higher than that of *Indica* (Fig. 4g; page 16, line 338-343), indicating that the genetic effect of dormancy was influenced by population structure.

Generally speaking, wild rice has strong dormancy. The growth period of wild rice varies greatly, so it is not easy to compare the germination rate at the same time, and some wild rice is vegetative reproduction. So, unfortunately, we did not obtain dormancy data for wild rice.

Page 15 line 298

It is suggested that the authors analyze whether *qSDR3.1* is selected during rice domestication, and then analyze the selection characteristics of chromosomal region carrying *qSDR3.1* gene.

Response: Thanks for the reviewer's suggestions. We analyzed that *qSDR3.1* was selected in the process of rice domestication, and the π value near *qSDR3.1* chromosome was lower than that around it. (Fig. 5d; Supplementary Fig. 22e-h; page 17, line 358-363)

EMI is an important downstream gene of the regulatory pathway of seed dormancy described in this paper. So, the authors need investigate the expression level of *EMI* in genetic materials and evaluate the seed dormancy of transgenic plants, including knock out and overexpression.

Response: Thanks for the reviewer's comments. We have investigated the expression level of *EMI* in genetic materials (Fig. 3b; page 13, line 266-270). Compared to ZH11, the germination rates of the *EMI* mutant lines were significantly higher, whereas the germination rate of the *EMI* overexpressed lines was significantly reduced (Supplementary Fig. 11d-e; page 12, line 247-250).

The findings in this study indicated that the *qSDR3.1* interacts with *ABI5*, regulating ABA signaling, thereby modulating seed dormancy in rice. So, the authors might investigate and compare the expression of ABA response genes and ABA content in ZH11, MJLXL, and transgenic plants.

Response: Thanks for the reviewer's comments. We have investigated and compared the expression of ABA response genes and ABA content in ZH11, MJLXL, and transgenic plants. The results showed that the expression levels of all detected ABA response genes in MJLXL, NIL, KO-1, and KO-2 with low germination rates were significantly higher than those in ZH11, COM-1, COM-2, and COM-3 with high germination rates (Fig. 3a-e; page 13, 266-270). The ABA content results showed that *qSDR3.1* regulated dormancy not by regulating ABA (Fig. 3f; page 13, line 275-277).

In the title, "*Oryza sativa*. L" should be "*Oryza sativa* L.". "Novel" should be removed.

Response: Thanks for the reviewer's careful observation. We have changed the title to "The quantitative trait locus *qSDR3.1* as a brake represses seed dormancy in rice". (page 1, line 1)

Page 6, line 124 and Figure 1d

The germination rate showed an obvious difference in the complementary transgenic

lines COM-1 (90%) and COM-2 (50%). Please investigate the seed dormancy of more than three Independent transgenic lines and explain the reason for the phenotypic difference of transgenic lines harboring the same construct.

Response: Thanks for the reviewer's comments. We have investigated the seed dormancy of three Independent transgenic lines (Fig. 1d, f). The copy number results showed that the copy number of COM-1 was almost twice that of COM-2 and COM-3 (Fig. 1e), and copy number variation would affect gene function. Therefore, we speculate that the significant difference in germination rate among COM-1, COM-2 and COM-3 is due to copy number variation. (page 7, line 143-144; page 19, line 403-408)
Page 11, line 212

The authors found that the knockout mutants of *qSDR3.1* showed a decrease of ABI5 protein content compared with the wild type. Please explain why the mutants with low ABI5 protein content had a strong transcription activity.

Response: Thanks for the reviewer's comments. We found that the protein band in the picture is not the target band, and the relevant description has been deleted in the MS.
Page 11, line 228

The authors did not describe the results of haplotype analysis in this paragraph.

Response: Thanks for the reviewer's careful observation. We have removed the "and haplotype" from the title and the results of haplotype analysis showed in page 14-15, line 305-311.

Page 14, line 292

The results of amino acid sequences analysis of wild rice are not included in this Supplementary Fig. 11.

Response: Thanks for the reviewer's careful observation. The meaning here refers to the wild rice in Supplementary Figure 14, and the amino acid analysis results are shown in Fig. 5a.

Page 19 line 405

"embryo" should be "shoot".

Response: Thanks for the reviewer's accurate description. "embryo" has been revised into "shoot". (page 23, line 498)

Fig. 1e

Please describe the genotype of this NIL.

Response: Thanks for the reviewer's comments. We have added the genotype description of NIL to the materials and methods. (page 23, line 488-490)

Figure S6

The germination rate of ZH11 had an obvious difference in panel a and b.

Response: Thanks for the reviewer's comments. The phenotype investigate times for panel a and b were different and have been marked in Supplementary Fig. 6.

Figure S7b

It is difficult to judge that the qSDR3.1-GFP signal was localized in the nucleus from these pictures. The nucleus-localized marker should be used in the subcellular location experiment.

Response: Thanks for the reviewer's comments. We have added a nuclear marker for experiments, and the results show that qSDR3.1-GFP signal was localized in the

nucleus. (Supplementary Fig. 7b; page 8-9, line 176-177).

Reviewer #4 (Remarks to the Author):

The results in MS "Novel quantitative trait locus *qSDR3.1* represses *Oryza sativa*. L seed dormancy" shown that discovery of *qSDR3.1*, a new quantitative trait locus (QTL) for seed dormancy in rice. knockout lines of *qSDR3.1* exhibited significant reduction in PHS. The *qSDR3.1* locus negatively regulated seed dormancy by directly inhibiting the transcriptional activity of ABSCISIC ACID INSENSITIVE5 (*ABI5*). All data are interesting. Although the *qSDR3.1* loci, *LOC_Os03g11550*, annotated as Mediator of OsbZIP46 Deactivation and Degradation(MODD), was reported which mediates the inactivation and degradation of OsbZIP46 and negatively regulates ABA signaling and drought resistance in rice, but the authors found its new function in seed dormancy.

Response: Thank you very much for reviewer's positive comments of this MS.

1. In Figure S1 and Figure S4, the seeds of MJLXL were not germinated at 7d, but the germinating rate of knockout mutant seeds reached 5%-10%. please discuss the results.

Response: Thanks for the reviewer's comments. On the one hand, the ABA content in MJLXL is significantly higher than that in ZH11 (Fig. 3f), indicating that the dormancy of MJLXL is not only influenced by ABA signals, but also by ABA content. On the other hand, in addition to *qSDR3.1*, there are two other QTLs that enhance the dormancy of MJLXL. This explains why MJLXL seeds hardly germinate on the seventh day (Supplementary Fig. 1a, b), but the germination rates of KO-1 and KO-1 reach 10-20% (Supplementary Fig. 4d, e). The dormancy mechanism of MJLXL and the molecular mechanism of *qSDR3.1* regulating dormancy have a more complex network, which requires further in-depth research in the future. (page 20-21, line 434-441)

2. The authors thought that the interaction between *qSDR3.1* and *ABI5* to regulate seed dormancy. In general, the *ABI5* protein in arabidopsis was not involved in seed dormancy. Please show the data and phenotype of rice *ABI5* mutant.

Response: Thanks for the reviewer's comments. We found that overexpression of *ABI5* significantly increased seed dormancy, whereas knocking out *ABI5* significantly reduced seed dormancy. (Supplementary Fig. 9; page 10-11, line 217-227)

3. To verify the relation of between *qSDR3.1* and *ABI5*, please construct these double mutants and analysis these mutant phenotype.

Response: Thanks for the reviewer's comments. To verify the relation of between *qSDR3.1* and *ABI5*, we construct double mutant in the ZH11 background (Supplementary Fig. 12a-b). Compared with ZH11, the germination rates of the double mutant was no significant difference (Supplementary Fig. 12c-d). These results further verified that *qSDR3.1* repressed the transcriptional activation of *ABI5* (page 12, line 260-263).

4. Figure S9 shown that the *ABI5* proteins binded with the promoter of *EM1*, but in many research works the *ABI5* was also binded with the promoter of *EM6*. Please check whether *ABI5* is able to bind the promoter of *EM6* to be involved in seed dormancy? and please check other genes involved in seed dormancy.

Response: Thanks for the reviewer's comments. *ABI5* binds to *EM6* in *Arabidopsis*

thaliana, but we did not find the *EM6* in rice. However, Through cut&Tag, Y1H, EMSA, and LUC experiments, we identified *LEA3* as another downstream marker gene of ABI5 to analyze the relationship between *qSDR3.1* and ABI5. (Supplementary Fig. 10d-f; page 11, line 238-241).

5. In introduction section, the authors said that seed dormancy was regulated by ABA and GA signaling, please check the ABA and GA contents of wild-type and mutants?

Response: Thanks for the reviewer's comments. We have examined the contents of ABA and GA in seeds, and the results showed that *qSDR3.1* regulated dormancy not by regulating ABA and GA content (Fig. 3f; Supplementary Fig. 13c-f; page 13, line 275-277).

Reviewers' Comments:

Reviewer #1:

Remarks to the Author:

The authors have addressed the most comments of the reviewers, and revised the manuscript accordingly. However, some concerns below still needs the authors consider.

1. The authors should provide the results of ABI5 expression in the NIL, COM and KO lines as vital evidence supporting that "The qSDR3.1 negatively regulated seed dormancy by directly inhibiting the transcriptional activity of ABIs" in abstract.
2. Suggest the authors to provide a diagram of the regulatory model of qSDR3.1 interacting with ABIs to regulate seed dormancy in the result or discussion part.
3. P6 lines 120-135, the authors mentioned "This region contains 4 candidate genes. After gene function prediction and sequencing analysis, we determined the candidate gene with MODD as qSDR3.1". It needs to be more rigorous to guess the candidate genes. There is no reason to exclude the other genes like LOC_Os03g11540, based solely on relevant reference and gene annotation. Some studies have reported that LOC_Os03g11540 encoding RPA1B, putative single-stranded DNA binding complex subunit 1, which is essential for transcriptional regulation and maintenance of genome stability. Additionally, the COM-1/2/3 lines do not fully complement the phenotype of NIL. It needs a further explanation.
4. Fig. 1c. What does the number under the gene model stand for? If it represents the position of nucleotide sequences of the gene, we suggest using the one relative to ATG ATG as 1. Fig. 4a has a similar question.
5. P13 lines 275-277, "In addition, we examined the contents of ABA and GA in seeds, and the results showed that qSDR3.1 regulated seed dormancy not by regulating ABA and GA content". This statement is inconsistent with Fig. 3f, which shows that the ABA content in NIL(qSDR 3.1) significantly differs from that in the knockout lines. Also, in Fig. S13c, there is a significant difference in the GA1 content between NIL(qSDR 3.1) and the knockout lines.
6. Figure S2b shows three dormancy QTLs detected on Chromosomes 2 and 3. However, we can not notice the small red line on the linkage map. Please remark on it.
7. The three reviewers all questioned the significant difference in germination rate (90.1% vs. 50.9%) among the complementary lines (COM-1, COM-2 and COM-3). The authors responded with the sentence "The copy numbers for COM-1, COM-2, and COM-3 were 13.15, 7.32 and 6.97, respectively (Fig. 1e)" in P7 Lines 143-144, and explained that the copy number variation of the gene may cause the difference (in lines 407-408). However, the copy number is not an integer, why? The authors should provide supporting evidence for that copy number. In addition, If qSDR3.1 interacts with ABI5 and regulates ABA signaling, why does Fig. 3 show no differences in the expression levels of signaling genes like LEA3 and EM1 among the three COM lines?
8. Line 371-372, "Hap5 had the weakest inhibitory effect (Fig. 24b)". It may be "Suppl fig 24b instead of "fig 24b" in this sentence.
9. P23 lines 488-489, It is unclear that "a single plant whose genotypes of all markers except P1 and P2 were consistent with ZH11 was selected as NIL". Please clarify how many markers are used to screen the background of NIL.
10. The quality of Figure S10s for EMSA results needs to be improved.
11. The data of two single mutants may be included in figure s12c for the clarity.

Reviewer #3:

Remarks to the Author:

This is my second review of this paper by Guo et al. I find that the authors have conducted additional experiments for addressing my prior concerns. Here are still some points for consideration.

1. The authors provided the genotypes of 108 BC3F1 plants in Supplementary Table 3. However, the

ratio of heterozygous and homozygous individuals did not fit to 1:1. In addition, the authors carried out QTL mapping using 153 lines derived from BC3F2 population. But the authors did not describe how to develop the mapping population. So, it is very difficult to determine whether the method of QTL mapping is accurate in this study.

2. The complementary transgenic plants (COM-1, COM-2, and COM-3) harboring qSDR3.1 from ZH11 showed a significant increase of germination rate compared with the control NIL, and the knockout mutants of qSDR3.1 gene had a significant decrease of germination rate compared with the wild type ZH11. This result indicated that the qSDR3.1 in ZH11 might be a gain of function allele. However, the authors did not provide the evidence for supporting that the qSDR3.1 in ZH11 might gain new function for seed dormancy in rice.

Reviewer #4:

The authors adequately responded to my comments and suggestions. But there are still some issues the authors would consider when revising the manuscript.

1. Please improve all picture definition, Figures and Supplemental figures are difficult to get information.
2. The method of Cut&Tag sequencing and analysis need describe more details, eg. How to harvest cells for releasing cell nuclei? the composition of wash buffer; the antibody? how to analyze the raw data... These are important for the results of the ABI5' target genes.
3. To verify the relation of between qSDR3.1 and ABI5, they constructed double mutant in the ZH11 (Supplementary Fig. 12a-b). But they only showed the phenotype of ZH11 and *qSDR3.1/ABI5* double mutant, the single mutant *qSDR3.1* and *abi5* also need be added, and these seeds are harvest at same time.
4. SDR3.1 interacts with ABI5 and represses the transcriptional activation function of ABI5, and SDR3.1 also inhibits OsABI3. Because ABI5 had an interaction with ABI3 (Lopez-Molina et al., 2002, PJ), and ABI3 also involve in seed dormancy, so SDR3.1 may participate in seed dormancy via OsABI5 and OsABI3. This possibility need to be proved, and the downstream genes may be deference if it would be confirmed.

Dear reviewers

Thank you very much for your time and comments on our manuscript. Those comments are all valuable and very helpful for getting the most out of our work. Our manuscript “Novel quantitative trait locus *qSDR3.1* represses *Oryza sativa*. L seed dormancy” Manuscript ID: NCOMMS-22-48784C, has been revised carefully and marked it in red and here is the listed response to the reviewers’ comments.

Thanks again for your time on our manuscript and any further comments from you and the four reviewers will be highly appreciated!

With best regards,

Sincerely yours,

Corresponding author: Zhonghua Sheng and Peisong Hu

First author: Naihui Guo

Response to reviewers:

Reviewer #1 (Remarks to the Author):

The authors have addressed the most comments of the reviewers, and revised the manuscript accordingly. However, some concerns below still needs the authors consider.

Response: We are pleased to have resolved most of the reviewer's comments. We are sorry that there are still some issues with the MS, so we would like to thank the reviewer once again for providing new comments to improve the quality of the MS. Additionally, since previous reviewer#2 is unavailable, thank you again for your additional comments.

1. The authors should provide the results of ABI5 expression in the NIL, COM and KO lines as vital evidence supporting that "The *qSDR3.1* negatively regulated seed dormancy by directly inhibiting the transcriptional activity of ABIs" in abstract.

Response: Thanks for the reviewer’s comments. We think what the reviewer wants to know is the expression level of ABI5 downstream genes (such as *EMI* and *LEA3*). "The *qSDR3.1* negatively regulated seed dormancy by directly inhibiting the transcriptional activity of ABIs" means that *qSDR3.1* inhibits the expression level of downstream genes of ABIs, rather than the expression level of ABIs themselves. Therefore, whether the expression levels of ABIs are different will not affect this conclusion. For this conclusion, firstly, we identified two downstream target genes of ABI5 (*EMI* and *LEA3*) (Supplementary Fig. 10). Then, LUC experiments demonstrated that *qSDR3.1* inhibits the activation of ABI5/ABI3 on both target genes (Fig. 2e-g; Supplementary Fig. 13a-

b). Finally, the expression levels of the two target genes were detected in genetic materials (Fig. 3a-b), and combined with the dormancy phenotypes of different materials (Supplementary Fig. 1; Fig. 1d; Supplementary Fig. 4e; Supplementary Fig. 9d), further demonstrating that "The *qSDR3.1* negatively regulated seed dormancy by directly inhibiting the transcriptional activity of ABIs". So, we have included the results of *EMI* and *LEA3* expression levels in abstract. (page 1-2, line 20-23).

2. Suggest the authors to provide a diagram of the regulatory model of *qSDR3.1* interacting with ABIs to regulate seed dormancy in the result or discussion part.

Response: Thanks for the reviewer's suggestion. We have provided a diagram of the regulatory model of *qSDR3.1* interacting with ABIs to regulate seed dormancy in the discussion (Supplementary Fig. 26). (page 21, line 439-441).

3. P6 lines 120-135, the authors mentioned "This region contains 4 candidate genes. After gene function prediction and sequencing analysis, we determined the candidate gene with MODD as *qSDR3.1*". It needs to be more rigorous to guess the candidate genes. There is no reason to exclude the other genes like LOC_Os03g11540, based solely on relevant reference and gene annotation. Some studies have reported that LOC_Os03g11540 encoding RPA1B, putative single-stranded DNA binding complex subunit 1, which is essential for transcriptional regulation and maintenance of genome stability. Additionally, the COM-1/2/3 lines do not fully complement the phenotype of NIL. It needs a further explanation.

Response: Thanks for the reviewer's comments. We fully agree that guessing candidate genes requires more rigor. For the 4 candidate genes, gene with unknown functions (LOC_Os03g11520) was generally excluded first (refer to Li, 2021. A genome-wide association study reveals that the 2-oxoglutarate/malate translocator mediates seed vigor in rice. Plant J). Then, the other three genes were sequenced, and the sequencing results showed no differences between the parents of the LOC_Os03g11530 and LOC_Os03g11540, while LOC_Os03g11550 had three base differences between the parents. Therefore, LOC_Os03g11550 is chosen as a candidate gene. (page 6-7, line 131-134).

As for the COM-1/2/3 lines do not fully complement the phenotype of NIL. According to the Supplementary Fig. 2d-e, *qSDR3.1* is an incomplete dominant gene. The *qSDR3.1* carried in NIL is the allele type of MJLXL. We have imported the *qSDR3.1* allele type in ZH11 into NIL, called COM-1/2/3. So COM-1/2/3 are heterozygous type of *qSDR3.1*, so their germination rate should be between NIL and ZH11, which can not fully complement the phenotype of NIL.

4. Fig. 1c. What does the number under the gene model stand for? If it represents the position of nucleotide sequences of the gene, we suggest using the one relative to ATG ATG as 1. Fig. 4a has a similar question.

Response: Thanks for the reviewer's suggestion. Yes, the number under the gene model represent the position of nucleotide sequences of the gene. We have made revisions in the Fig. 1c and Fig. 4a.

5. P13 lines 275-277, "In addition, we examined the contents of ABA and GA in seeds, and the results showed that *qSDR3.1* regulated seed dormancy not by regulating ABA and GA content". This statement is inconsistent with Fig. 3f, which shows that the ABA

content in NIL(*qSDR 3.1*) significantly differs from that in the knockout lines. Also, in Fig. S13c, there is a significant difference in the GA1 content between NIL(*qSDR 3.1*) and the knockout lines.

Response: Thanks for the reviewer's comments. Generally speaking, ABA promotes seed dormancy while GA inhibits seed dormancy. Compared with NIL, the ABA content of COM lines significantly increased (Fig. 3f), which is inconsistent with the significantly higher germination rate of COM compared to NIL (Fig. 1d). Similarly, compared to ZH11, the ABA content of the KO lines significantly decreased (Fig. 3f), which is inconsistent with the significantly lower germination rate of the KO line compared to ZH11 (Supplementary Fig. 4e). Therefore, ABA is not the reason why *qSDR3.1* regulates dormancy.

As for in Fig. S13c. Although the GA1 content in the KO lines were significantly reduced compared to ZH11, the GA1 content in the COM lines did not significantly increase compared to NIL. It is difficult to explain that *qSDR3.1* regulates dormancy through GA1, and based on the measurement results of GA3, GA4, and GA7 (Supplementary Fig. 13d-f), we believe that *qSDR3.1* also does not regulate dormancy through GA content.

6. Figure S2b shows three dormancy QTLs detected on Chromosomes 2 and 3. However, we can not notice the small red line on the linkage map. Please remark on it.

Response: Thanks for the reviewer's comments. For clarity, we have added red arrows (Supplementary Fig. 2b).

7. The three reviewers all questioned the significant difference in germination rate (90.1% vs. 50.9%) among the complementary lines (COM-1, COM-2 and COM-3). The authors responded with the sentence "The copy numbers for COM-1, COM-2, and COM-3 were 13.15, 7.32 and 6.97, respectively (Fig. 1e)" in P7 Lines 143-144, and explained that the copy number variation of the gene may cause the difference (in lines 407-408). However, the copy number is not an integer, why? The authors should provide supporting evidence for that copy number. In addition, If *qSDR3.1* interacts with *ABI5* and regulates ABA signaling, why does Fig. 3 show no differences in the expression levels of signaling genes like *LEA3* and *EMI* among the three COM lines?

Response: Thanks for the reviewer's comments. Because we used RT-PCR to detect the copy number (page 24-25, line 524-527), the value is not an integer, but the copy number should indeed be an integer. Therefore, we adjusted the copy number of the complementary lines to an integer (page 7, line 149; Fig. 1e).

Why does Fig. 3 show no differences in the expression levels of signaling genes like *LEA3* and *EMI* among the three COM lines? That's a good question. Generally speaking, if the germination rate of COM-1 is higher than that of -2 and -3, then its expression levels of *EMI* and *LEA3* should be lower than those of -2 and -3. We speculate that if the copy number of COM-1 is higher, it is more likely that it will insert into the chromosome and disrupt the function of other genes. Changes in ABA signaling, such as decreased expression levels of *EMI* and *LEA3*, are not the only reason for the increase in COM-1 germination rate. Other dormancy regulating genes are disrupted by copy number insertion. These factors together result in a higher germination rate of COM-1 compared to -2 and -3.

8. Line 371-372, "Hap5 had the weakest inhibitory effect (Fig. 24b)". It may be "Suppl fig 24b instead of "fig 24b" in this sentence.

Response: Thanks for the reviewer's careful observation. It is "Supplementary Fig. 24" instead of "Fig. 24". We have already conducted it. (page 18, line 373,377).

9. P23 lines 488-489, It is unclear that "a single plant whose genotypes of all markers except P1 and P2 were consistent with ZH11 was selected as NIL". Please clarify how many markers are used to screen the background of NIL.

Response: Thanks for the reviewer's comments. 108 pairs of SSR polymorphic markers evenly distributed on 12 chromosomes were used for background screening of NIL. (page 23, line 495-496).

10. The quality of Figure S10s for EMSA results needs to be improved.

Response: Thanks for the reviewer's comments. We have improved the results of EMSA (Supplementary Fig. 10).

11. The data of two single mutants may be included in figure s12c for the clarity.

Response: Thanks for the reviewer's comments. We have added the germination phenotype of two single mutants in Supplementary Fig. 12.

Reviewer #3 (Remarks to the Author):

This is my second review of this paper by Guo et al. I find that the authors have conducted additional experiments for addressing my prior concerns. Here are still some points for consideration.

Response: We are pleased to have resolved the reviewer's prior concerns. We are sorry that there are still some issues with the MS, so we would like to thank the reviewer once again for providing new comments to improve the quality of the MS.

1. The authors provided the genotypes of 108 BC₃F₁ plants in Supplementary Table 3. However, the ratio of heterozygous and homozygous individuals did not fit to 1:1. In addition, the authors carried out QTL mapping using 153 lines derived from BC₃F₂ population. But the authors did not describe how to develop the mapping population. So, it is very difficult to determine whether the method of QTL mapping is accurate in this study.

Response: Thanks for the reviewer's comments. The genotype of BC₃F₁ selected for localization is shown in Supplementary Table 3, instead of the genotype of 108 BC₃F₁ plants. The value 108 represents the number of markers rather than the number of individual plants. The 153 BC₃F₂ plants used for localization were self bred from BC₃F₁ genotype in Supplementary Table 3. (page 5, line 107-108).

2. The complementary transgenic plants (COM-1, COM-2, and COM-3) harboring *qSDR3.1* from ZH11 showed a significant increase of germination rate compared with the control NIL, and the knockout mutants of *qSDR3.1* gene had a significant decrease of germination rate compared with the wild type ZH11. This result indicated that the *qSDR3.1* in ZH11 might be a gain of function allele. However, the authors did not provide the evidence for supporting that the *qSDR3.1* in ZH11 might gain new function for seed dormancy in rice.

Response: Thanks for the reviewer's comments. In the introduction, we introduced that seed dormancy is a domesticated trait (page 2, line40-43). Wild rice has strong seed dormancy, and the most combination of aa¹⁰² and aa¹⁵⁶ was tested as Hap5 type (Fig. 5a-b). With artificial domestication, Hap1 and Hap2 types, mainly composed of *indica* (Fig. 4d), and Hap3 and Hap4 types, mainly composed of *japonica* (Fig. 4d), were differentiated. The molecular mechanism of *qSDR3.1* regulating dormancy is achieved by inhibiting the transcriptional activation activity of ABIs, thereby inhibiting ABA signaling (Fig. 2f-g; Fig. 3a-e; Supplementary Fig. 13b). The LUC results showed that Hap3 and Hap4 had the strongest inhibitory effect on ABI5 transcription activity, Hap1 and Hap2 had the second inhibitory effect, and Hap5 had the weakest inhibitory effect (Supplementary Fig. 24b). The above results indicate that the inhibitory effect of *qSDR3.1* on ABIs is enhanced from wild rice to cultivated rice. This can be understood as *qSDR3.1* in ZH11 (*japonica*), which has a stronger effect than *qSDR3.1* in wild rice, i.e. obtaining a new function - inhibiting seed dormancy.

Reviewer #4 (Remarks to the Author):

The authors adequately responded to my comments and suggestions. But there are still some issues the authors would consider when revising the manuscript.

Response: We are pleased to have adequately resolved the reviewer's comments and suggestions. We are sorry that there are still some issues with the MS, so we would like to thank the reviewer once again for providing new comments and suggestions to improve the quality of the MS.

1. Please improve all picture definition, Figures and Supplemental figures are difficult to get information.

Response: Thanks for the reviewer's comments. To reduce memory, images placed in word will be compressed, resulting in unclear images. We have merged the Figures and Supplemental figures into PDFs.

2. The method of Cut&Tag sequencing and analysis need describe more details, eg. How to harvest cells for releasing cell nuclei? the composition of wash buffer; the antibody? how to analyze the raw data... These are important for the results of the ABI5' target genes.

Response: Thanks for the reviewer's comments. We have provided detailed information in the corresponding position of the MS. The cell nucleus was extracted by using CellLytic TM PN Isolation/Extraction Kit (Weibo:3424255, refer to its manual for specific steps), wash buffer come from the Kit; primary antibody (Mouse anti GFP-Tag mAb, ABclonal, AE012, dilution 1:100); second antibody (HRP-conjugated Mouse anti GFP-Tag mAb, ABclonal, AE030, dilution 1:100); Evaluated the raw data using FastQC software (v0.11.5). Then, the Trimmomatic software (v0.39) was used for data filtering, and the adapter sequence at the end was truncated. Next, bwa software (v0.7.17-r1188) was used to perform unique comparison, screening and de duplication processing on the obtained reads. Finally, MACS2 software (v2.2.7.1) was used to find the protein binding sites (peaks). (page 29, line 623-635).

3. To verify the relation of between *qSDR3.1* and ABI5, they constructed double mutant in the ZH11 (Supplementary Fig. 12a-b). But they only showed the phenotype of ZH11

and *qSDR3.1/ABI5* double mutant, the single mutant *qSDR3.1* and *abi5* also need be added, and these seeds are harvest at same time.

Response: Thanks for the reviewer's comments. We have added the germination phenotype of two single mutants in Supplementary Fig. 12.

4. *SDR3.1* interacts with *ABI5* and represses the transcriptional activation function of *ABI5*, and *SDR3.1* also inhibits *OsABI3*. Because *ABI5* had an interaction with *ABI3* (Lopez-Molina et al., 2002, PJ), and *ABI3* also involve in seed dormancy, so *SDR3.1* may participate in seed dormancy via *OsABI5* and *OsABI3*. This possibility need to be proved, and the downstream genes may be deference if it would be confirmed.

Response: Thanks for the reviewer's comments. Yes, *qSDR3.1* participate in seed dormancy via *ABIs* (*ABI5* and *ABI3*). In Figures 2f and g, *qSDR3.1* inhibits the activation effect of *ABI5* on *EM1* and *LEA3*; In Figure 13b, *qSDR3.1* inhibits the activation effect of *ABI3* on *EM1* and *LEA3*. Knocking out *qSDR3.1* results in a decrease in germination rate (Supplementary Fig. 4e), indicating that *qSDR3.1* inhibits seed dormancy. Knocking out *ABI5* showed an increase in germination rate (Supplementary Fig. 9d), while knocking out *ABI3* also showed an increase in germination rate (Chen, 2021, The Crop Journal, *OsVP1* activates *Sdr4* expression to control rice seed dormancy via the ABA signaling pathway), indicating that *ABI5* and *ABI3* promote seed dormancy. The above result meant the *qSDR3.1* negatively regulated seed dormancy by directly inhibiting the transcriptional activity of *ABIs* (*ABI5* and *ABI3*).

Reviewers' Comments:

Reviewer #1:

Remarks to the Author:

The authors have addressed my concerns in the revised manuscript. I have no further comments on the revision.

Reviewer #3:

Remarks to the Author:

Authors all addressed my questions and concerns and revised the relevant statements appropriately. I believe this manuscript provided valuable scientific and useful agricultural information.

Reviewer #4:

Remarks to the Author:

The authors adequately responded to my comments and suggestions.

Response to reviewers:

Reviewer #1 (Remarks to the Author):

The authors have addressed my concerns in the revised manuscript. I have no further comments on the revision.

Response: Thank you very much.

Reviewer #3 (Remarks to the Author):

Authors all addressed my questions and concerns and revised the relevant statements appropriately. I believe this manuscript provided valuable scientific and useful agricultural information.

Response: Thank you very much.

Reviewer #4 (Remarks to the Author):

The authors adequately responded to my comments and suggestions.

Response: Thank you very much.